# ErbB4 deletion in noradrenergic neurons in the locus coeruleus induces mania-like behavior via elevated catecholamines

**Shu-Xia Cao[1†], Ying Zhang[2†], Xing-Yue Hu[1], Bin Hong[1], Peng Sun[2], Hai-Yang He[2], Hong-Yan Geng[2], Ai-Min Bao[2], Shu-Min Duan[2], Jian-Ming Yang[3], Tian-Ming Gao[3], Hong Lian[2]*, Xiao-Ming Li[1,2]***

[1]Sir Run Run Shaw Hospital, Zhejiang University School of Medicine, Hangzhou, China; [2]Center for Neuroscience, Key Laboratory of Medical Neurobiology of the Ministry of Health of China, Zhejiang University School of Medicine, Hangzhou, China; [3]Department of Neurobiology, School of Basic Medical Sciences, Southern Medical University, Guangzhou, China

**Abstract** Dysfunction of the noradrenergic (NE) neurons is implicated in the pathogenesis of bipolar disorder (BPD). ErbB4 is highly expressed in NE neurons, and its genetic variation has been linked to BPD; however, how ErbB4 regulates NE neuronal function and contributes to BPD pathogenesis is unclear. Here we find that conditional deletion of ErbB4 in locus coeruleus (LC) NE neurons increases neuronal spontaneous firing through NMDA receptor hyperfunction, and elevates catecholamines in the cerebrospinal fluid (CSF). Furthermore, *Erbb4*-deficient mice present mania-like behaviors, including hyperactivity, reduced anxiety and depression, and increased sucrose preference. These behaviors are completely rescued by the anti-manic drug lithium or antagonists of catecholaminergic receptors. Our study demonstrates the critical role of ErbB4 signaling in regulating LC-NE neuronal function, reinforcing the view that dysfunction of the NE system may contribute to the pathogenesis of mania-associated disorder.
DOI: https://doi.org/10.7554/eLife.39907.001

**\*For correspondence:**
honglian@zju.edu.cn (HL);
lixm@zju.edu.cn (X-ML)

[†]These authors contributed equally to this work

**Competing interests:** The authors declare that no competing interests exist.

## Introduction

Bipolar disorder (BPD), diagnosed on the basis of manic episodes with or without depression, is a severely debilitating psychiatric disorder (*Holden, 2008*). Though risk genes and rodent models of BPD have been reported (*Arey et al., 2014*; *Craddock and Sklar, 2009*; *Gouvea et al., 2016*; *Han et al., 2013*; *Roybal et al., 2007*; *Saul et al., 2012*), the underlying pathogenic mechanism has not yet been clearly defined due to the phenotypic and genotypic complexity of this disorder (*Harrison et al., 2018*).

Several lines of evidence implicate the noradrenergic (NE) system in the pathology of BPD. For instance, the concentrations of norepinephrine and its metabolites are significantly upregulated in the cerebrospinal fluid (CSF) of BPD patients during the manic state (*Kurita, 2016*; *Manji et al., 2003*; *Post et al., 1973*; *Post et al., 1978*). In contrast, norepinephrine is downregulated in patients with depressive disorder (*Maas et al., 1971*; *Moret and Briley, 2011*; *Wiste et al., 2008*) and associated with mood transition in BPD patients (*Kurita, 2016*; *Salvadore et al., 2010*). However, how the NE system is involved in the pathology of BPD remains uncertain.

ErbB4, a receptor tyrosine kinase, plays a vital role in a number of biological processes, including neural development, excitability, and synaptic plasticity (*Mei and Nave, 2014*). In parvalbumin-positive (PV) interneurons, ErbB4 is involved in the etiology of schizophrenia and epilepsy (*Chen et al., 2010*; *Del Pino et al., 2013*; *Fisahn et al., 2009*; *Kx et al., 2012*; *Tan et al., 2011*). ErbB4 mRNA is

**eLife digest** Bipolar disorder is a mental illness that affects roughly 1 in 100 people worldwide. It features periods of depression interspersed with episodes of mania – a state of delusion, heightened excitation and increased activity. Evidence suggests that changes in a brain region called the locus coeruleus contribute to bipolar disorder. Cells within this area produce a chemical called norepinephrine, whose levels increase during mania and decrease during depression. But it is unclear exactly how norepinephrine-producing cells, also known as noradrenergic cells, contribute to bipolar disorder.

The answer may lie in a protein called ErbB4, which is found within the outer membrane of many noradrenergic neurons. ErbB4 is active in both the developing and adult brain, and certain people with bipolar disorder have mutations in the gene that codes for the protein. Might changes in ErbB4 disrupt the activity of noradrenergic neurons? And could these changes increase the risk of bipolar disorder?

To find out, Cao, Zhang et al. deleted the gene for ErbB4 from noradrenergic neurons in the locus coeruleus of mice. The mutant mice showed mania-like behaviors: compared to normal animals, they were hyperactive, less anxious, and consumed more of a sugary solution. Treating the mice with lithium, a medication used in bipolar disorder, reversed these changes and made the rodents behave more like non-mutant animals. Further experiments revealed that noradrenergic neurons in the mutant mice showed increased spontaneous activity. These animals also had more of the chemicals noradrenaline and dopamine in the fluid circulating around their brains and spinal cords.

The results thus suggest that losing ErbB4 enhances the spontaneous firing of noradrenergic neurons in the locus coeruleus. This increases release of noradrenaline and dopamine, which in turn leads to mania-like behaviors. Future research should examine whether drugs that target ErbB4 could treat mania and improve the lives of people with bipolar disorder and related conditions.
DOI: https://doi.org/10.7554/eLife.39907.002

also prominently expressed in locus coeruleus (LC) NE neurons(*Gerecke et al., 2001*), and coding variants of *ERBB4* are genetically associated with BPD susceptibility (*Chen et al., 2012*; *Bipolar Genome Study et al., 2011*). However, how ErbB4 regulates NE neuronal function and whether NE neuron-specific ErbB4 signaling participates in the pathogenesis of BPD is unknown.

In this study, we achieved ErbB4 deletion primarily in NE neurons by crossing *Th-Cre* mice (*Gelman et al., 2003*), in which Cre recombinase is mainly expressed in NE neurons of the LC (see *Figure 1A–E*, *Figure 1—figure supplement 1* and Discussion section), with mice carrying the loxP-flanked *Erbb4* allele (*Erbb4$^{loxp/loxp}$*). ErbB4 deletion increases the spontaneous firing of LC-NE neurons in an NMDA receptor-dependent manner, and elevates the concentrations of norepinephrine and dopamine in the CSF. Furthermore, *Th-Cre;Erbb4$^{loxp/loxp}$* mice manifest a mania-like behavioral profile that can be recapitulated by *Erbb4$^{loxp/loxp}$* mice with region-specific ablation of ErbB4 in the LC. In addition, treatment with lithium, a commonly used clinical anti-manic drug, or antagonists against dopamine or norepinephrine receptors all rescue the mania-like behaviors in *Th-Cre;Erbb4$^{loxp/loxp}$* mice. Taken together, our study linked ErbB4 physiological function with NE system homeostasis and demonstrated the pathogenic effect of ErbB4 dysregulation in NE neurons in mania-associated psychiatric diseases.

## Results

### ErbB4 is primarily deleted from LC-NE neurons in *Th-Cre;Erbb4$^{loxp/loxp}$* mice

To determine Cre distribution in our specific *Th-Cre* mouse line, we crossed *Th-Cre* mice with Ai9 mice to label Cre-positive neurons with red fluorescent protein tdTomato (*Madisen et al., 2010*). We examined Cre expression in the LC, ventral tegmental nucleus (VTA), and substantia nigra pars compacta (SNC) of Ai9;*Th-Cre* mice at postnatal day (P) 50 because tyrosine hydroxylase (TH), the key enzyme for the synthesis of norepinephrine and dopamine, is mainly expressed in these three

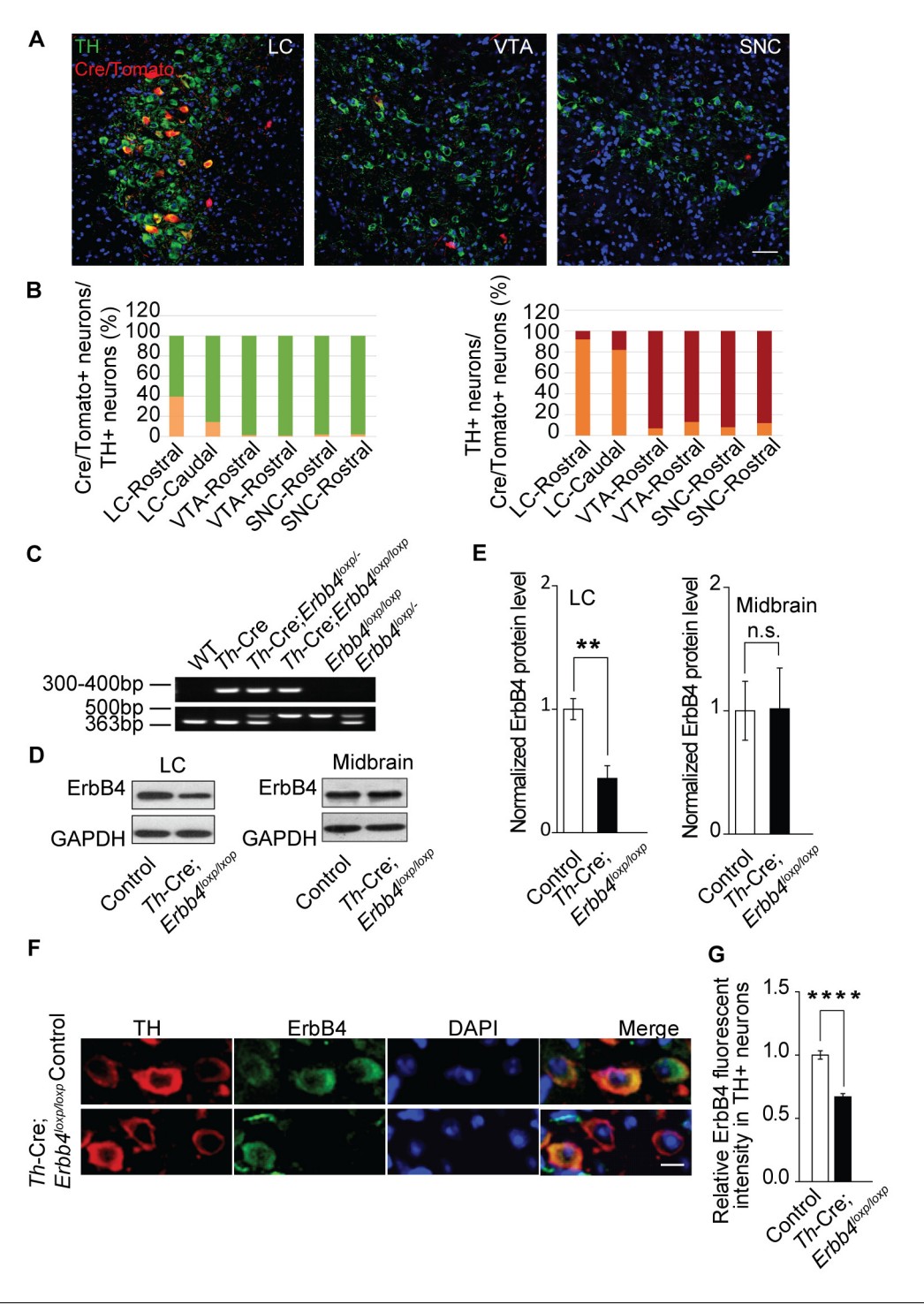

**Figure 1.** ErbB4 is primarily deleted in NE neurons of the LC in *Th-Cre;Erbb4^{loxp/loxp}* mice. (**A**) Representative micrographs of Cre/Tomato distribution (red) in the locus coeruleus (LC), ventral tegmental area (VTA), and substantia nigra pars compacta (SNC). Slices were obtained from Ai9;*Th-Cre* mice and stained with antibody to TH (green), a marker of NE and dopaminergic neurons. Scale bar, 50 μm. (**B**) Colocalization of TH and Cre/Tomato. Three mice were studied, with three slices chosen for each mouse. (**C**) Genotyping of *Th-Cre;Erbb4^{loxp/loxp}* mice. The *Erbb4* primers generated a 363-base pair (bp) product for the wild-type allele or a 500 bp product for the loxP-flanked allele. The *Th-Cre* primers generated a band between 300 and 400 bp. (**D**), (**E**) Quantification of the fold change in ErbB4 protein expression relative to control mice. Unpaired two-tailed Student's t-test. Data are
*Figure 1 continued on next page*

*Figure 1 continued*

expressed as means ± s.e.m. **p<0.01. (**F**) Specific deletion of ErbB4 in NE neurons of the LC. Sections from *Th-Cre* mice and *Th-Cre;Erbb4$^{loxp/loxp}$* mice were stained with ErbB4-specific antibody and TH-specific antibody. Sections were also stained with DAPI to indicate nuclei. Scale bar, 10 μm. (**G**) Quantification of ErbB4 fluorescent intensity in TH-positive (TH+) cells. n = 13 slices (control), random 16 – 30 TH+ cells were quantified from each slice. n = 11 slices (*Th-Cre;Erbb4$^{loxp/loxp}$*), random 19 – 30 TH+ cells were quantified from each slice.

DOI: https://doi.org/10.7554/eLife.39907.003

The following source data and figure supplements are available for figure 1:

**Source data 1.** Statistical reporting of *Figure 1*.
DOI: https://doi.org/10.7554/eLife.39907.007
**Figure supplement 1.** Cre/GFP was primarily expressed in NE neurons of the LC in Ai3;*Th-Cre* mice.
DOI: https://doi.org/10.7554/eLife.39907.004
**Figure supplement 2.** ErbB4 was primarily deleted in the LC of *Th-Cre;Erbb4$^{loxp/loxp}$* mice.
DOI: https://doi.org/10.7554/eLife.39907.005
**Figure supplement 3.** No obvious differences were detected in cell density or soma size of LC neurons between control and *Th-Cre;Erbb4$^{loxp/loxp}$* mice.
DOI: https://doi.org/10.7554/eLife.39907.006

areas. Colocalization analysis of TH staining and tdTomato suggested that in the rostral part of the LC, approximately 40% of TH-positive (TH+) neurons were Cre/Tomato-positive (Cre/Tomato+) and 92% of Cre/Tomato+ neurons were TH+, whereas in the caudal part of the LC, Cre/Tomato+ neurons only constituted 14% of TH+ neurons, with 82% of Cre/Tomato+ neurons being TH+ (*Figure 1A,B*). The VTA and SNC contain approximately 70% of the dopaminergic neurons in the brain (*Björklund and Dunnett, 2007*). Unexpectedly, in contrast to the LC, there were very few Cre/Tomato+ neurons in the VTA and SNC. Cre/Tomato was only expressed in approximately 1.6% and 0.9% of neurons in the rostral and caudal VTA, respectively. Moreover, only 2.1% of neurons in the rostral and caudal parts of the SNC were Cre/Tomato+. In the rostral and caudal parts of the VTA and SNC, only 8% and 12% of Cre/Tomato+ neurons, respectively, were TH+ (*Figure 1A,B*). To exclude possible false-positive signals introduced by the reporter mouse line, we took advantage of Ai3 mice, another reporter mouse strain that labels Cre-positive neurons with green fluorescent protein (GFP), to confirm these results. Consistently, we observed very little Cre expression in the VTA or SNC (*Figure 1—figure supplement 1*). These data suggest that Cre recombinase was primarily expressed in the NE neurons of the LC in our *Th-Cre* mouse line.

To investigate its role in NE neurons, we deleted ErbB4 in NE neurons by crossing *Th-Cre* with *Erbb4$^{loxp/loxp}$* mice (*Figure 1C*). Immunoblotting analysis showed that ErbB4 was significantly decreased in the LC of *Th-Cre;Erbb4$^{loxp/loxp}$* mice (*Figure 1D,E*, and *Figure 1—figure supplement 2*) with no significant change in the midbrain (VTA and SNC) (*Figure 1D,E*), which is consistent with our previous observation that Cre was mainly expressed in LC-NE neurons in *Th-Cre* mice. Immunohistochemical analysis also confirmed the deletion of ErbB4 in the LC (*Figure 1F,G*). In addition, we observed no obvious changes in cell density or soma size of LC neurons in *Th-Cre;Erbb4$^{loxp/loxp}$* mice compared to control mice (*Figure 1F* and *Figure 1—figure supplement 3*).

## Hyperactive LC-NE neurons in *Th-Cre;Erbb4$^{loxp/loxp}$* mice increase extracellular norepinephrine and dopamine

To analyze the influence of ErbB4 deficiency on LC-NE neuronal physiology, we measured the spontaneous activity of LC-NE neurons in *Th-Cre;Erbb4$^{loxp/loxp}$* mice in cell-attached configuration. Consistent with previous studies, LC-NE neurons recorded from slices of the control mice exhibited a firing rate of 1.94 ± 0.18 Hz (*Chandler et al., 2014*; *Jedema and Grace, 2004*) (*Figure 2A,C*). However, the spontaneous firing rate of LC-NE neurons in the *Th-Cre;Erbb4$^{loxp/loxp}$* mice was significantly increased (2.87 ± 0.18 Hz) (*Figure 2B,C*), and the inter-spike interval was significantly decreased (*Figure 2D*).

Previous studies have demonstrated that neuronal excitability can affect the expression and phosphorylation of TH, the rate limiting factor in catecholamine synthesis (*Aumann et al., 2011*; *Chevalier et al., 2008*; *Lew et al., 1999*; *Zigmond et al., 1989*). Using protein extracts of LC tissues from controls and mutants, we measured the expression of phosphorylated TH (TH-Ser40), an active

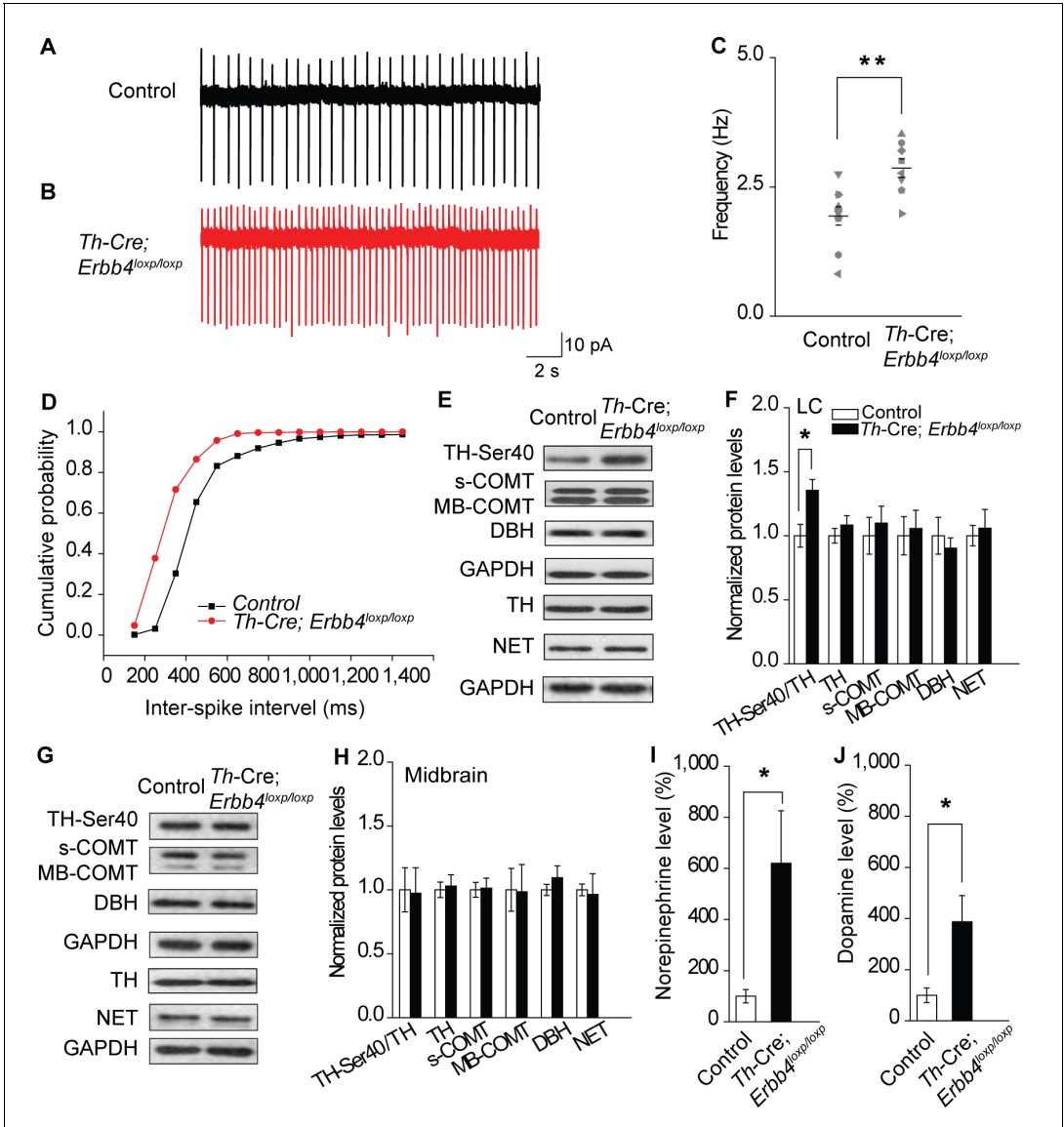

**Figure 2.** Increased spontaneous firing of LC-NE neurons, extracellular norepinephrine, and intracellular TH phosphorylation in *Th-Cre;Erbb4^{loxp/loxp}* mice. (A), (B) Representative firing of LC-NE neurons from control (black) and *Th-Cre;Erbb4^{loxp/loxp}* mice (red). (C) Spontaneous firing frequency of LC-NE neurons was increased in *Th-Cre;Erbb4^{loxp/loxp}* mice. n = 10 from three mice (control); n = 8 from three mice (*Th-Cre;Erbb4^{loxp/loxp}*). (D) Interspike intervals were calculated over 2 min of firing from each neuron. Interspike intervals were decreased in *Th-Cre;Erbb4^{loxp/loxp}* mice compared with control mice. n = 10 from three mice (control); n = 8 from three mice (*Th-Cre;Erbb4^{loxp/loxp}*). Two-sample Kolmogorov-Smirnov test and data in (D) are presented in a cumulative frequency plot. ****p<0.0001. (E), (F) Protein levels of TH-Ser40 were increased in the LC of *Th-Cre;Erbb4^{loxp/loxp}* mice. NET, norepinephrine reuptake transporter; DBH, dopamine beta-hydroxylase; S-COMT, soluble catechol-o-methyltransferase; MB-COMT, membrane-binding form of COMT. (G), (H) No significant change was detected in the dopaminergic neurons clustered in the midbrain (VTA and SNC). (I), (J) In vivo microdialysis and HPLC data suggested that norepinephrine and dopamine levels were significantly increased in *Th-Cre;Erbb4^{loxp/loxp}* mice. Standard curves are presented in *Figure 2—figure supplement 2*. n = 6 mice (control); n = 6 mice (*Th-Cre;Erbb4^{loxp/loxp}*). Unpaired two-tailed Student's t-test. Data are expressed as means ± s.e.m. *p<0.05.

DOI: https://doi.org/10.7554/eLife.39907.008

The following source data and figure supplements are available for figure 2:

**Source data 1.** Statistical reporting of *Figure 2*.
DOI: https://doi.org/10.7554/eLife.39907.011

**Figure supplement 1.** Representative Western blots of TH and COMT in the LC of control and *Th-Cre;Erbb4^{loxp/loxp}* mice.
DOI: https://doi.org/10.7554/eLife.39907.009

**Figure supplement 2.** HPLC analysis of norepinephrine and dopamine.

*Figure 2 continued on next page*

*Figure 2 continued*

DOI: https://doi.org/10.7554/eLife.39907.010

form of TH required for norepinephrine synthesis, along with other enzymes involved in norepineph-rine homeostasis, including dopamine beta-hydroxylase (DBH), norepinephrine transporter (NET), and catechol-O-methyltransferase (COMT). We observed a marked increase in TH-Ser40 but not in total TH (*Figure 2E,F*, and *Figure 2—figure supplement 1*). In contrast, DBH, another enzyme involved in norepinephrine synthesis, and NET and COMT, which regulate norepinephrine degrada-tion, were unchanged (*Figure 2E,F*). Thus, changes in the neuronal excitability of LC-NE neurons in *Th-Cre;Erbb4^{loxp/loxp}* mice may specifically increase TH phosphorylation. Using lysates from the mid-brain, none of these proteins showed any changes (*Figure 2G,H*), suggesting region-specific norepi-nephrine synthetic activity influenced in mutant animals.

As LC-NE neurons are the major source of norepinephrine in the forebrain (*Sara, 2009*), we hypothesized that the increase in NE neuronal and TH activities in the LC might increase the level of norepinephrine in the brain. Therefore, we examined the norepinephrine level in *Th-Cre;Erbb4^{loxp/loxp}* mice using in vivo microdialysis in the lateral ventricle of anaesthetized mice, followed by high-performance liquid chromatography (HPLC). Results showed that norepinephrine concentration was significantly increased in the CSF of *Th-Cre;Erbb4^{loxp/loxp}* mice (*Figure 2I* and *Figure 2—figure sup-plement 2*). Given that dopamine, the precursor of norepinephrine, is coupled with changes in nor-epinephrine level and can be co-released with norepinephrine by NE neurons (*Devoto et al., 2005*; *Guiard et al., 2008*; *Pozzi et al., 1994*; *Yamamoto and Novotney, 1998*), we also examined the concentration of dopamine in the CSF in *Th-Cre;Erbb4^{loxp/loxp}* mice. Remarkably, the concentration of dopamine was also obviously increased compared with that in the control mice (*Figure 2J* and *Figure 2—figure supplement 2*).

Taken together, the increased excitability of LC-NE neurons may increase TH phosphorylation, resulting in the increase of norepinephrine and dopamine observed in the CSF of *Th-Cre;Erbb4^{loxp/loxp}* mice.

## Increased spontaneous firing of LC-NE neurons in *Th-Cre;Erbb4^{loxp/loxp}* mice due to NMDA receptor hyperfunction

An increase in glutamatergic synaptic input (*Jodo and Aston-Jones, 1997*; *Somogyi and Llewellyn-Smith, 2001*), or decrease in feedback inhibition from α-2-adrenoceptor, an autoreceptor (*Langer, 1980*; *Starke, 2001*), may contribute to the increased firing rate of LC-NE neurons. More-over, in studies on the hippocampus and prefrontal cortex, the NMDA receptor (NMDAR), especially its subunit isoform NR2B, is reported to be regulated by ErbB4 (*Hahn et al., 2006*; *Pitcher et al., 2011*). Therefore, we examined the expression of NMDAR subunits (NR2B, NR1, and NR2A) and autoreceptors α-2A (A2A) and α-2C (A2C) using protein samples from the LC of the controls and *Th-Cre;Erbb4^{loxp/loxp}* mice. Results showed that the expression of NR2B was significantly increased in *Th-Cre;Erbb4^{loxp/loxp}* mice, whereas no changes were detected in the expressions of NR1, A2A, A2C or NR2A (*Figure 3A,B*).

NR2B overexpression may alter NMDAR activity (*Galliano et al., 2018*). Therefore, we recorded evoked NMDAR-mediated current of LC-NE neurons in acute brain slices from *Ai9;Th-Cre* mice and *Ai9;Th-Cre;Erbb4^{loxp/loxp}* mice and compared the current amplitude. NMDAR-mediated current showed significantly increased amplitude in Ai9; *Th-Cre;Erbb4^{loxp/loxp}* mice compared with Ai9; *Th-Cre* mice (*Figure 3C,D*), thus indicating NMDAR hyperfunction in the LC-NE neurons in the absence of ErbB4. Meanwhile, to test whether the balance between excitatory synapses and inhibitory synap-ses and the intrinsic excitability of the LC-NE neurons were altered by ErbB4 deletion, we recorded spontaneous EPSC (sEPSC) and spontaneous IPSC (sIPSC) of LC-NE neurons from the control *Ai9; Th-Cre* mice and ErbB4-deficient *Ai9;Th-Cre;Erbb4^{loxp/loxp}* mice. Neither sEPSC nor sIPSC presented any changes in their amplitude or frequency (*Figure 3—figure supplement 1*). In addition, intrinsic properties of LC-NE neurons measured by analysis of action potential (AP) threshold, AP amplitude, AP half-width, afterhyperpolarization (AHP), rheo-based current, Cm, Rin, and τ were unchanged compared with those of the control mice (*Figure 3—figure supplement 2*).

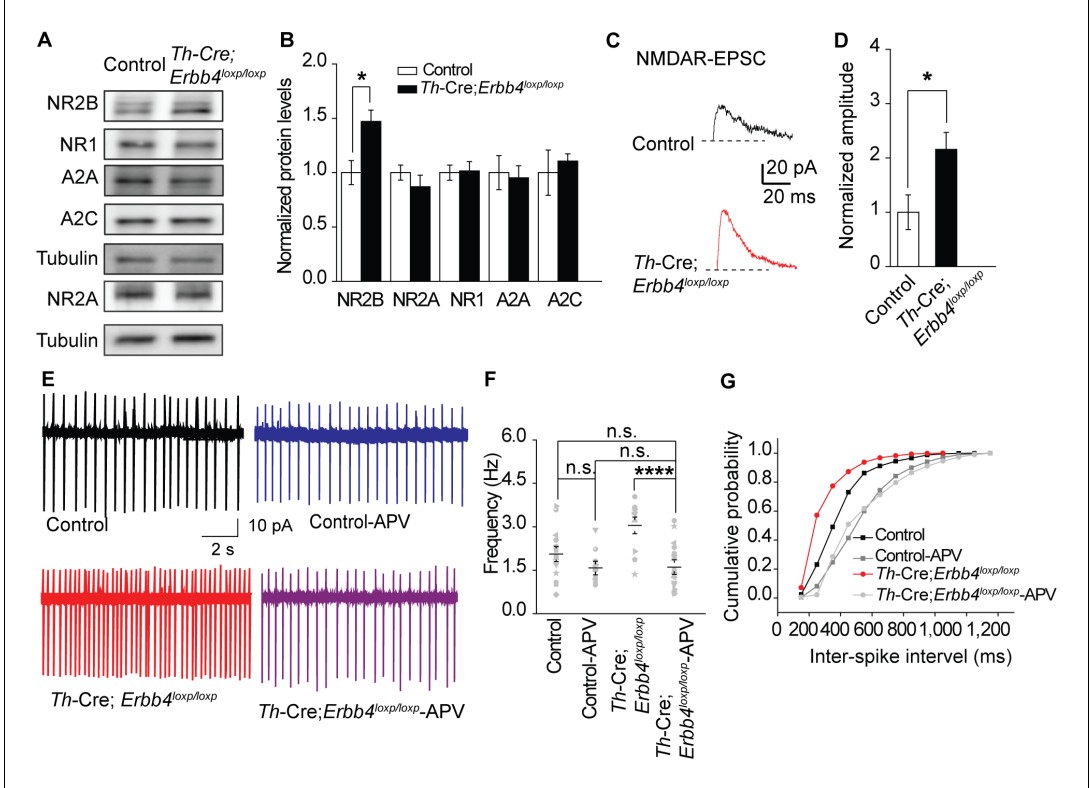

**Figure 3.** NMDA receptor mediates hyperexcitability of LC-NE neurons in *Th-Cre;Erbb4*[loxp/loxp] mice. (**A**), (**B**) Protein levels of NMDA receptor subunit NR2B were increased in the LC of *Th-Cre;Erbb4*[loxp/loxp] mice. Unpaired two-tailed Student's t-test. Data are expressed as means ± s.e.m. *p<0.05. (**C**) Representative NMDAR-EPSC current traces from LC-NE neurons in *Ai9;Th-Cre* (control) and *Ai9; Th-Cre;Erbb4*[loxp/loxp] mice. (**D**) Amplitude of NMDAR current recorded in LC-NE neurons is significantly increased in ErbB4-deficient mice compared with control mice. n = 7 from three mice (control); n = 8 from three mice (*Th-Cre;Erbb4*[loxp/loxp]). (**E**) Representative firing of LC-NE neurons from control and *Th-Cre;Erbb4*[loxp/loxp] mice untreated or treated with APV (50 μM), an NMDA receptor antagonist. (**F**) Spontaneous firing frequency of LC-NE neurons was rescued by APV (50 μM) in *Th-Cre;Erbb4*[loxp/loxp] mice. n = 13 from three mice (*Th-Cre;Erbb4*[loxp/loxp]); n = 20 from three mice (*Th-Cre;Erbb4*[loxp/loxp] + APV). Two-way ANOVA. Data are expressed as means ± s.e.m. *p<0.05. (**G**) Inter-spike intervals were significantly rescued by APV (50 μM) in *Th-Cre;Erbb4*[loxp/loxp] mice. n = 13 from three mice (*Th-Cre;Erbb4*[loxp/loxp]); n = 20 from three mice (*Th-Cre;Erbb4*[loxp/loxp] + APV). Two-sample Kolmogorov-Smirnov test and data in (**G**) are presented in a cumulative frequency plot. ***p<0.001.

DOI: https://doi.org/10.7554/eLife.39907.012
The following source data and figure supplements are available for figure 3:

**Source data 1.** Statistical reporting of *Figure 3*.
DOI: https://doi.org/10.7554/eLife.39907.015
**Figure supplement 1.** Spontaneous excitation and inhibition balance measured by sEPSC and sIPSC is not changed in LC-NE neurons of *Th-Cre; Erbb4*[loxp/lox] mice.
DOI: https://doi.org/10.7554/eLife.39907.013
**Figure supplement 2.** Intrinsic properties of LC-NE neurons are unchanged in *Th-Cre; ErbB4*[loxp/loxp] mice.
DOI: https://doi.org/10.7554/eLife.39907.014

We hypothesized that the increase in NE neuronal activity might be attributed to strengthened NMDAR function in *Th-Cre;Erbb4*[loxp/loxp] mice. Using the patch clamp technique, we found that the spontaneous firing rates and inter-spike intervals of LC-NE neurons in *Th-Cre;Erbb4*[loxp/loxp] mice were rescued by the NMDAR antagonist APV (50 μM) (*Figure 3E–G*). Thus, NMDARs appear to mediate the hyperexcitability of LC-NE neurons in *Th-Cre;Erbb4*[loxp/loxp] mice.

## *Th-Cre;Erbb4*[loxp/loxp] mice show mania-like behaviors

The LC is involved in mood, reward, and motor ability (*Borodovitsyna et al., 2017*; *Bouret and Sara, 2005*; *Sara, 2009*). As ErbB4 was mainly deleted from the LC-NE neurons in *Th-Cre;Erbb4*[loxp/loxp] mice, we hypothesized that *Th-Cre;Erbb4*[loxp/loxp] mice might exhibit LC-related behavioral

abnormalities. We first examined the motor ability of *Th-Cre;Erbb4*<sup>loxp/loxp</sup> mice using the open field test. *Th-Cre;Erbb4*<sup>loxp/loxp</sup> mice traveled longer distances and at higher speeds than control (*Erb-B4*<sup>loxp/loxp</sup>) mice (*Figure 4A–E*) and spent less time immobile (*Figure 4F*). To examine anxiety- and depression-related behaviors of *Th-Cre;Erbb4*<sup>loxp/loxp</sup> mice, we conducted the elevated plus maze test (EPM) and forced swim test. In the EPM, *Th-Cre;Erbb4*<sup>loxp/loxp</sup> mice spent more time in and

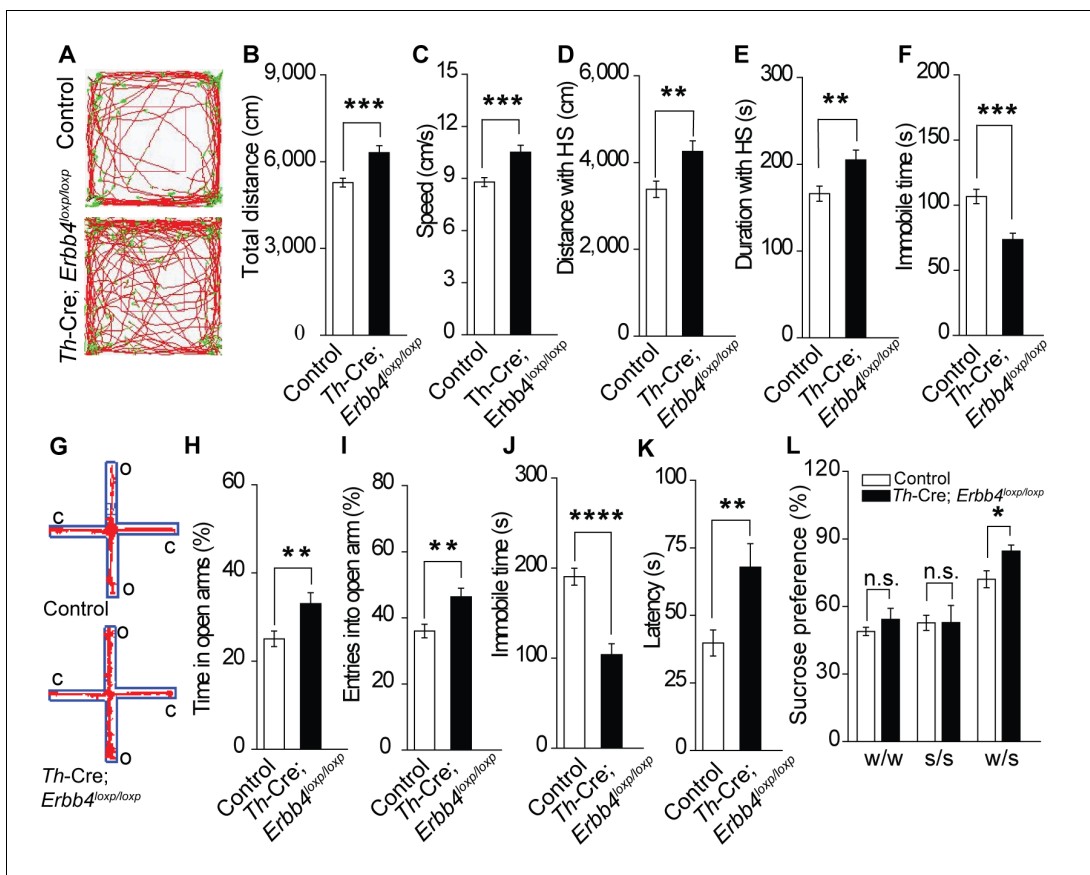

**Figure 4.** *Th-Cre;Erbb4*<sup>loxp/loxp</sup> mice show mania-like behaviors. (A) Representative trajectories of control (*Erbb4*<sup>loxp/loxp</sup>) and *Th-Cre;Erbb4*<sup>loxp/loxp</sup> mice. We defined high speed (red line) as >10 cm/s, immobility as <2 cm/s, and low speed (green line) as 2 – 10 cm/s. (B), (C) Locomotor activity (B) and speed (C) of control and *Th-Cre; Erbb4*<sup>loxp/loxp</sup> mice in open field tests. n = 24 (control); n = 22 (*Th-Cre;Erbb4*<sup>loxp/loxp</sup>). (D), (E) Distance (D) and duration (E) traveled at high speed (HS). n = 17 (control); n = 13 (*Th-Cre;Erbb4*<sup>loxp/loxp</sup>). (F) Immobility time during open field tests decreased in *Th-Cre;Erbb4*<sup>loxp/loxp</sup> mice. (G) Examples of the performance of control and *Th-Cre; Erbb4*<sup>loxp/loxp</sup> mice in the EPM test. C, closed arm; O, open arm. (H), (I) Performance of control and *Th-Cre; Erbb4*<sup>loxp/loxp</sup> mice in the EPM test. n = 34 (control); n = 24 (*Th-Cre;Erbb4*<sup>loxp/loxp</sup>). (J), (K) Immobility time (J) and latency to first surrender (K) in the forced swim test. n = 22 (control); n = 13 (*Th-Cre;Erbb4*<sup>loxp/loxp</sup>). (L) Sucrose preference of control and *Th-Cre;Erbb4*<sup>loxp/loxp</sup> mice. Water (w). Sucrose (s). n = 19 (control); n = 14 (*Th-Cre; Erbb4*<sup>loxp/loxp</sup>). Unpaired two-tailed Student's t-test. Data are expressed as means ± s.e.m. *p<0.05, **p<0.01, ***p<0.001, ****p<0.0001. n.s., not significant.

DOI: https://doi.org/10.7554/eLife.39907.016

The following source data and figure supplements are available for figure 4:

**Source data 1.** Statistical reporting of *Figure 4*.
DOI: https://doi.org/10.7554/eLife.39907.019

**Figure supplement 1.** There was no significant difference in body weight between control and *Th-Cre;Erbb4*<sup>loxp/loxp</sup> mice and no deficit of *Th-Cre;Erbb4*<sup>loxp/loxp</sup> mice in the prepulse inhibition experiment.
DOI: https://doi.org/10.7554/eLife.39907.017

**Figure supplement 2.** No significant change in the distance travelled in center and time spent in center area between control (*Erbb4*<sup>loxp/loxp</sup>) mice and *Th-Cre; Erbb4*<sup>loxp/loxp</sup> mice in open field test.
DOI: https://doi.org/10.7554/eLife.39907.018

presented more entries into the open arms compared with the control mice (*Figure 4G–I*). In the forced swim test, *Th-Cre;Erbb4^{loxp/loxp}* mice showed less immobility (*Figure 4J*) and longer latency to first surrender compared with the control mice (*Figure 4K*). To examine the responses of *Th-Cre; Erbb4^{loxp/loxp}* mice to a natural reward, we performed the sucrose preference test. During this test, *Th-Cre;Erbb4^{loxp/loxp}* mice displayed increased preference for sucrose compared with the control mice (*Figure 4L*). No significant deficits in body weight or prepulse inhibition were observed in *Th-Cre;Erbb4^{loxp/loxp}* mice (*Figure 4—figure supplement 1*). In addition, no significant change was detected between control mice and *Th-Cre; ErbB4^{loxp/loxp}* mice in the distance travelled in center and time spent in center area in open field test (*Figure 4—figure supplement 2*). These data indicate that *Th-Cre;Erbb4^{loxp/loxp}* mice exhibited hyperactivity, decreased anxiety and depression, and increased sucrose preference, thus resembling the phenotypes of rodent mania models (*Arey et al., 2014*; *Cosgrove et al., 2016*; *Han et al., 2013*; *Kirshenbaum et al., 2011*; *Nestler and Hyman, 2010*; *Prickaerts et al., 2006*; *Roybal et al., 2007*; *Shaltiel et al., 2008*).

## Viral-mediated LC-specific ErbB4-deficient mice recapitulates mania-like behaviors and molecular and electrophysiological changes of *Th-Cre; Erbb4^{loxp/loxp}* mice

To exclude the possibility that the *Th-Cre;Erbb4^{loxp/loxp}* mouse phenotypes were attributed to Cre-expressing neurons in other brain areas, we tested whether region-specific deletion of ErbB4 in the LC was sufficient to induce mania-like behaviors. We injected an adeno-associated virus (AAV) expressing Cre and GFP (AAV-Cre-GFP) into the LC of *Erbb4^{loxp/loxp}* mice bilaterally. Cre/GFP was expressed abundantly in the LC (*Figure 5A*), with 52% of LC neurons being Cre/GFP-positive (Cre/GFP+) and 77.7% of Cre/GFP + neurons being TH+ (*Figure 5—figure supplement 1*). Of the left 22.3% Cre/GFP+ but TH-negative (TH-) neurons, 55.5% among them showed ErbB4 expression (*Figure 5—figure supplement 2*). Using immunoblotting, we confirmed the efficiency of ErbB4 deletion in the LC of *Erbb4^{loxp/loxp}* mice after AAV-Cre-GFP injection (*Figure 5B*). Behavioral tests were carried out 4 weeks after viral injection. In the open field test, viral-mediated region-specific deletion of ErbB4 in the LC significantly increased locomotor activity and traveling speed (*Figure 5C,D*), and both the distance and time traveled at high speed were significantly increased (*Figure 5E,F*). Moreover, the immobility time in the open area was significantly reduced (*Figure 5G*). In the EPM, ErbB4 deletion in the LC significantly increased both the time in and number of entries into the open arms (*Figure 5H,I*). In the forced swim test, mice with LC ErbB4 deletion exhibited decreased immobility time and increased latency to first surrender (*Figure 5J,K*). In the sucrose preference test, LC ErbB4 deletion significantly increased sucrose preference of the injected mice (*Figure 5L*).

In addition to behavioral performance, AAV-mediated LC-specific ErbB4 deletion mice showed similar molecular and electrophysiological abnormalities to *Th-Cre;Erbb4^{loxp/loxp}* mice. Spontaneous excitability (*Figure 6A–C*), NR2B expression (*Figure 6D,E*), and NMDAR-mediated current amplitude (*Figure 6F,G*) were increased in LC-NE neurons infected by AAV-Cre-GFP viruses in *Erbb4^{loxp/loxp}* mice. In summary, the results showed that region-specific deletion of ErbB4 in the LC is sufficient to induce similar electrophysiological, biochemical, and mania-like behavioral phenotypes as those manifested in *Th-Cre;Erbb4^{loxp/loxp}* mice, thus indicating that ErbB4 deletion in LC-NE neurons plays the prominent role in abnormalities of *Th-Cre;Erbb4^{loxp/loxp}* mice as opposed to potential off-target ErbB4 deletions in non-LC regions.

## Lithium treatment of *Th-Cre;Erbb4^{loxp/loxp}* mice rescues behavioral, molecular, and electrophysiological abnormalities

Lithium was the first medicine approved by the Food and Drug Administration for BPD treatment. Testing the effect of lithium treatment on the behavioral abnormalities of *Th-Cre;Erbb4^{loxp/loxp}* mice may validate those behaviors to be mania-like and also reveal whether NE neurons are involved in the mechanism of lithium.

After mice were treated for 10 d with lithium chloride (600 mg L$^{-1}$) dissolved in drinking water, as described previously (*Roybal et al., 2007*), the behavioral performance of *Th-Cre;Erbb4^{loxp/loxp}* mice was rescued in the open field test, EPM, forced swim test, and sucrose preference test (*Figure 7*). In the open field test, lithium decreased locomotor activity, traveling speed, and distance and time traveled at high speed, and increased immobility time of *Th-Cre;Erbb4^{loxp/loxp}* mice (*Figure 7A–E*).

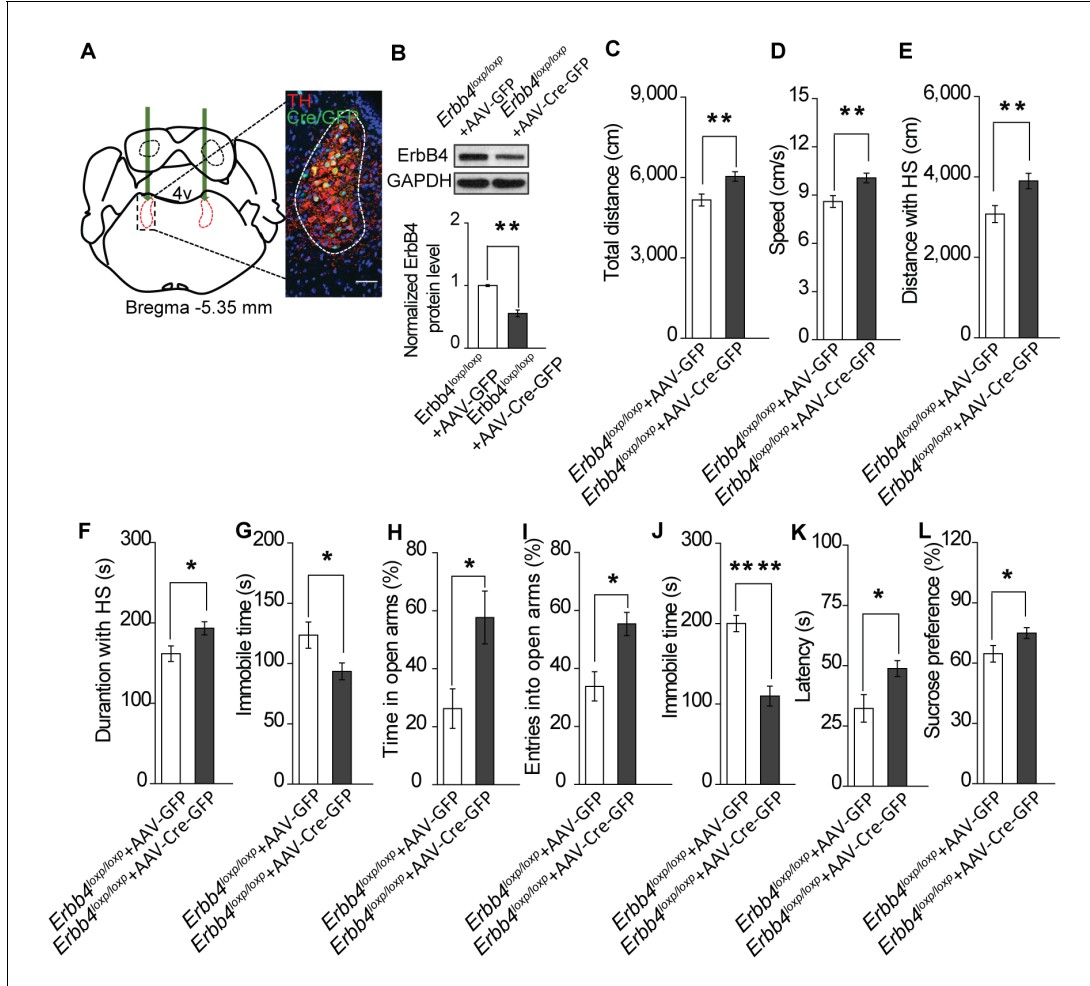

**Figure 5.** Specific ablation of ErbB4 in the LC is sufficient to cause mania-like behaviors. (A) Illustration of bilateral viral injection of AAV-Cre-GFP in the mouse LC. LC sections were examined for Cre/GFP (green) 5 weeks after stereotaxic microinjection of AAV-Cre-GFP into the LC of *Erbb4^{loxp/loxp}* mice; antibody staining for TH is shown in red. Scale bars, 50 μm. Cartogram is presented in *Figure 5—figure supplement 1* . (B) ErbB4 expression detected by immunoblotting was significantly decreased in LC protein lysates from *Erbb4^{loxp/loxp}* mice after AAV-Cre-GFP injection. n = 4 (AAV-GFP); n = 4 (AAV-Cre-GFP). (C, D) Locomotor activity (C) and speed (D) of mice injected with AAV-GFP or AAV-Cre-GFP in the open field test. n = 14 (AAV-GFP); n = 12 (AAV-Cre-GFP). (E–G) Distance (E) and duration (F) traveled at HS and immobility time (G) of mice in the open field test after viral injection. n = 14 (AAV-GFP); n = 12 (AAV-Cre-GFP). (H, I) Percentage of time (H) and entries (I) into the open arms by mice injected with AAV-GFP or AAV-Cre-GFP in the EPM test. n = 10 (AAV-GFP); n = 18 (AAV-Cre-GFP). (J, K) Immobility time (J) and latency to first surrender (K) in the forced swim test with AAV-GFP or AAV-Cre-GFP injection. n = 11 (AAV-GFP); n = 18 (AAV-Cre-GFP). (L) Sucrose preference of mice injected with AAV-GFP or AAV-Cre-GFP. n = 10 (AAV-GFP); n = 13 (AAV-Cre-GFP). Unpaired two-tailed Student's t-test. Data are expressed as means ± s.e.m. *p<0.05, **p<0.01, ****p<0.0001.
DOI: https://doi.org/10.7554/eLife.39907.020

The following source data and figure supplements are available for figure 5:

**Source data 1.** Statistical reporting of *Figure 5*.
DOI: https://doi.org/10.7554/eLife.39907.025

**Figure supplement 1.** Cartogram of the colocalization of Cre/GFP-positive (Cre/GFP+) and NE neurons (TH+) in the LC 5 weeks after viral injection.
DOI: https://doi.org/10.7554/eLife.39907.021

**Figure supplement 2.** 55.5% of TH-GFP + neurons in AAV-injected *Erbb4^{loxp/loxp}* mice are ErbB4+.
DOI: https://doi.org/10.7554/eLife.39907.022

**Figure supplement 3.** ErbB4-deficient mice are not an ADHD model.
DOI: https://doi.org/10.7554/eLife.39907.023

**Figure supplement 4.** Normal CREB signaling activity in *Th-Cre;ErbB4^{loxp/loxp}* mice.
DOI: https://doi.org/10.7554/eLife.39907.024

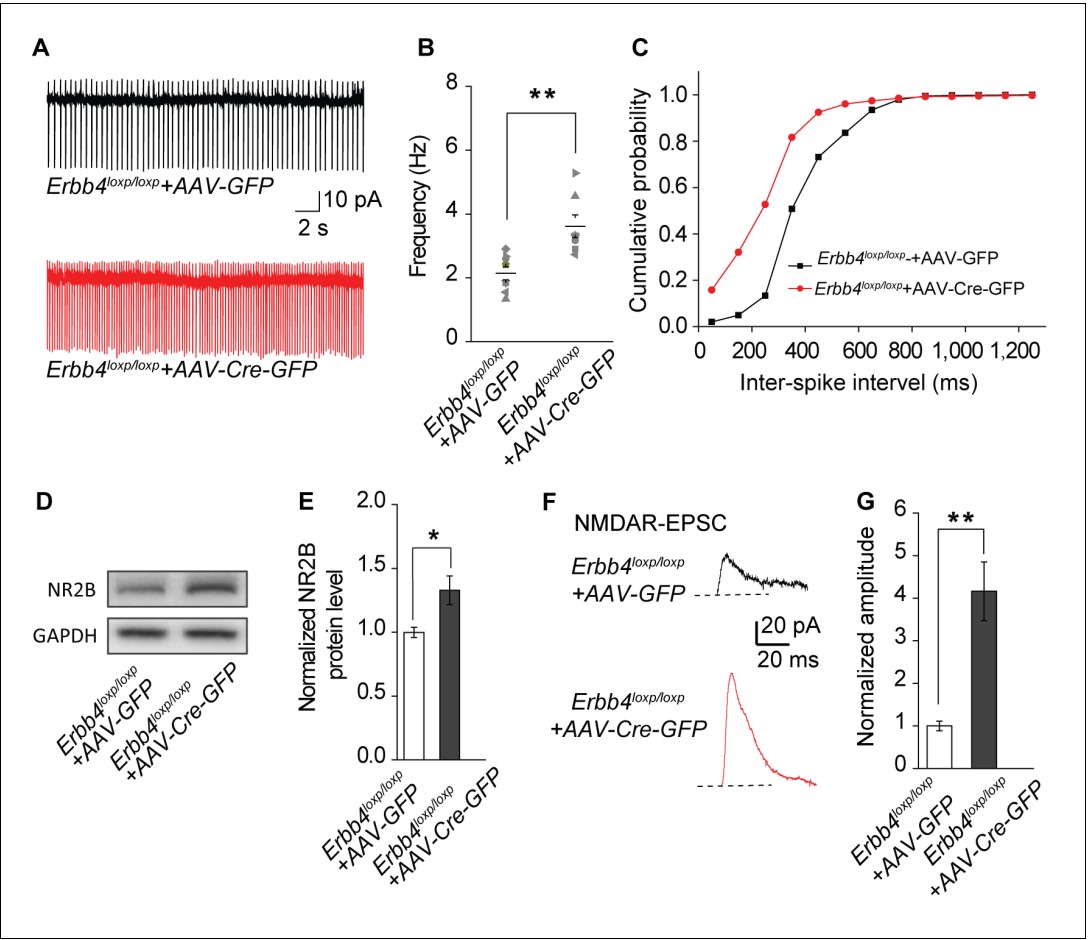

**Figure 6.** The spontaneous excitability, NR2B expression, and NMDAR current of LC-NE neurons are increased in *ErbB4^loxp/loxp^* mice bilaterally injected with AAV-Cre-GFP virus into the LC. (**A**) Representative spontaneous firing traces of LC-NE neurons in *ErbB4^loxp/loxp^* mice bilaterally injected with AAV- GFP or AAV-Cre-GFP virus. (**B**), (**C**) Quantification of firing frequency (**B**) and cumulative histogram of inter-spike interval (**C**) of spontaneous firing. n = 8 (*ErbB4^loxp/loxp^*-AAV-GFP); n = 7 (*ErbB4^loxp/loxp^*-AAV-Cre-GFP). (**D**), (**E**) Representative blot (**D**) and quantification (**E**) of NR2B protein level from LC protein lysates of *ErbB4^loxp/loxp^* mice received AAV-GFP or AAV-Cre-GFP injection. n = 3 (control); n = 3 (*Th-Cre; ErbB4^loxp/loxp^*). (**F**), (**G**) NMDAR current is enhanced in *ErbB4^loxp/loxp^* mice bilaterally injected with AAV-Cre-GFP. n = 3 (*ErbB4^loxp/loxp^*-AAV-GFP); n = 3 (*ErbB4^loxp/loxp^*-AAV-Cre-GFP). Unpaired two-tailed Student's t-test and Two-sample Kolmogorov-Smirnov test. Data are expressed as means ± s.e.m. *p<0.05, **p<0.01.

DOI: https://doi.org/10.7554/eLife.39907.026

The following source data is available for figure 6:

**Source data 1.** Statistical reporting of *Figure 6*.
DOI: https://doi.org/10.7554/eLife.39907.027

In addition, lithium decreased both time in and entries into the open arms by *Th-Cre;Erbb4^loxp/loxp^* mice in the EPM (*Figure 7F–G*). The treated *Th-Cre;Erbb4^loxp/loxp^* mice also exhibited significantly increased immobility time and decreased latency to first surrender in the forced swim test (*Figure 7H,I*), and reduced sucrose preference in the sucrose preference test (*Figure 7J*).

To better understand the mechanisms underlying the effect of lithium on mania-like behaviors of *Th-Cre;Erbb4^loxp/loxp^* mice, Western blotting and patch clamp recordings were performed to detect NR2B expression, TH phosphorylation, and spontaneous firing of LC-NE neurons after lithium treatment. The NR2B protein level and phosphorylation of TH were significantly decreased in *Th-Cre;Erbb4^loxp/loxp^* mice receiving lithium (*Figure 7K,L*), whereas no change was observed in the protein levels of TH or the membrane-bound and soluble forms of COMT (MB-COMT and S-COMT,

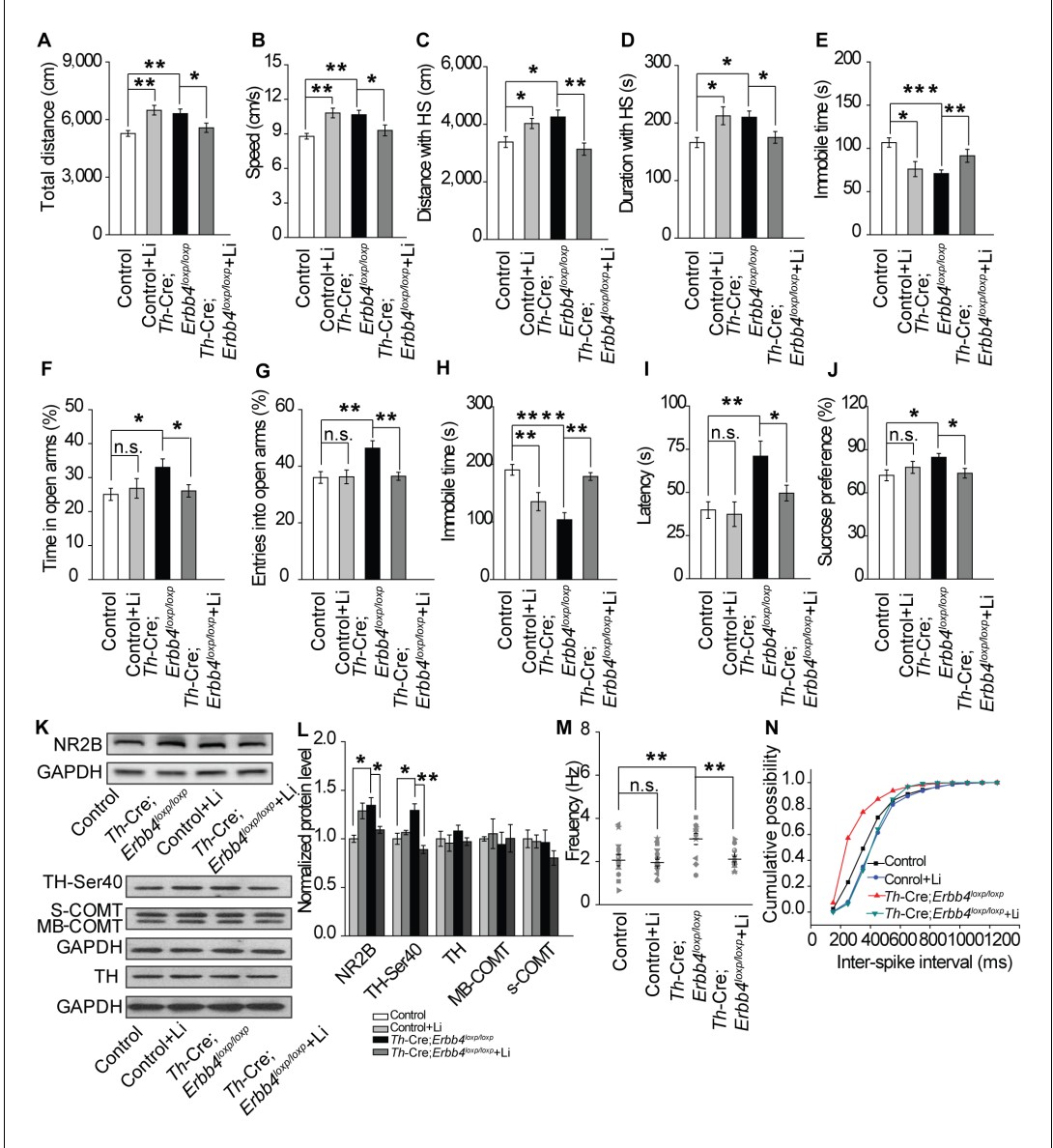

**Figure 7.** Lithium rescued the behavioral, molecular, and electrophysiological phenotypes of *Th-Cre;Erbb4^loxp/loxp^* mice. (A), (B) Locomotor activity (A) and speed (B) in the open field test. n = 24 (*Erbb4^loxp/loxp^*), n = 10 (*Erbb4^loxp/loxp^* + lithium), n = 22 (*Th-Cre;Erbb4^loxp/loxp^*), n = 13 (*Th-Cre;Erbb4^loxp/loxp^* + lithium). (C–E) Distance (C) and duration (D) traveled at HS and immobility time (E) after lithium treatment. (F), (G) Percentage of time (F) and entries (G) into open arms by *Th-Cre;Erbb4^loxp/loxp^* mice with and without lithium in the EPM test. n = 23 (*Erbb4^loxp/loxp^*); n = 8 (*Erbb4^loxp/loxp^* + lithium); n = 24 (*Th-Cre;Erbb4^loxp/loxp^*); n = 18 (*Th-Cre;Erbb4^loxp/loxp^* + lithium). (H), (I) Immobility time (H) and latency to first surrender (I) in forced swim test. n = 20 (*Erbb4^loxp/loxp^*); n = 8 (*Erbb4^loxp/loxp^* + lithium); n = 16 (*Th-Cre;Erbb4^loxp/loxp^*); n = 20 (*Th-Cre;Erbb4^loxp/loxp^* + lithium). (J) Sucrose preference of *Th-Cre;Erbb4^loxp/loxp^* mice treated with lithium. n = 19 (*Erbb4^loxp/loxp^*); n = 12 (*Erbb4^loxp/loxp^* + lithium); n = 13 (*Th-Cre;Erbb4^loxp/loxp^*); n = 16 (*Th-Cre; Erbb4^loxp/loxp^* + lithium). (K) Western blots of LC samples from *Th-Cre;Erbb4^loxp/loxp^* mice with and without lithium treatment. (L) Protein level of NR2B and TH-Ser40 in the LC after lithium treatment. Protein levels of TH, s-COMT, and MB-COMT were not significantly changed in the LC after lithium treatment. TH, tyrosine hydroxylase; S-COMT, soluble catechol-o-methyltransferase; MB-COMT, membrane-binding form of COMT. n = 4 (*Erbb4^loxp/loxp^*); n = 4 (*Erbb4^loxp/loxp^* + lithium); n = 4 (*Th-Cre;Erbb4^loxp/loxp^*); n = 4 (*Th-Cre;Erbb4^loxp/loxp^* + lithium). (M), Spontaneous firing of LC-NE neurons after lithium treatment. n = 13 from three mice (*Th-Cre;Erbb4^loxp/loxp^*); n = 15 from three mice (*Th-Cre;Erbb4^loxp/loxp^* + lithium). (N) Interspike intervals after lithium treatment. n = 13 from three mice (*Th-Cre;Erbb4^loxp/loxp^*); n = 15 from three (*Th-Cre;Erbb4^loxp/loxp^* + lithium). Two-way ANOVA. Data are expressed as means ± s.e.m. Two-sample Kolmogorov-Smirnov test and data in (N) are presented as a cumulative frequency plot. *p<0.05, **p<0.01, ****p<0.0001. n.s., not significant.

DOI: https://doi.org/10.7554/eLife.39907.028

The following source data is available for figure 7:

**Source data 1.** Statistical reporting of *Figure 7*.
DOI: https://doi.org/10.7554/eLife.39907.029

respectively) (*Figure 7K,L*). In addition, the spontaneous firing rates and inter-spike intervals of LC-NE neurons in *Th-Cre;Erbb4loxp/loxp* mice were both rescued after lithium treatment (*Figure 7M,N*).

These results indicate that NE neurons may be a potential target of lithium in the treatment of mania. In addition, the rescuing effect of lithium on the behavioral abnormalities of *Th-Cre;Erbb4loxp/loxp* mice further indicated that the behaviors induced by ErbB4 deletion in LC-NE neurons are mostly mania-like phenotypes.

### Increase in norepinephrine and dopamine contributes to mania-like behaviors in *Th-Cre;Erbb4loxp/loxp* mice

Our previous results showed that both norepinephrine and dopamine were increased in *Th-Cre;Erbb4loxp/loxp* mice (*Figure 2I,J*). To identify which system contributes to the mania-like behaviors of *Th-Cre;Erbb4loxp/loxp* mice, norepinephrine α1 receptor antagonist prazosin (1 mg/kg, i.p.) and dopamine D1 receptor antagonist SCH23390 (0.125 mg/kg, i.p.) were used to inhibit the effects of norepinephrine and dopamine, respectively. Both prazosin and SCH23390 decreased the locomotor activity and traveling speed of *Th-Cre;Erbb4loxp/loxp* mice in the open field test (*Figure 8A,B*). Furthermore, both distance and time traveled at high speed decreased (*Figure 8C,D*), and the immobility time was markedly increased (*Figure 8E*) after prazosin and SCH23390 treatment. In the EPM test, prazosin and SCH23390 treatment in *Th-Cre;Erbb4loxp/loxp* mice decreased the time spent in the open arms, although no effect on the number of entries was observed (*Figure 8F,G*). Moreover, *Th-Cre;Erbb4loxp/loxp* mice treated with prazosin or SCH23390 exhibited increased immobility time and decreased latency to first surrender in the forced swim test (*Figure 8H,I*). In the sucrose preference test, prazosin and SCH23390 both significantly decreased the sucrose preference of *Th-Cre;Erbb4loxp/loxp* mice (*Figure 8J*). Similar to their effect on *Th-Cre;Erbb4loxp/loxp* mice, prazosin and SCH23390 reduced locomotor activity, elevated depression and anxiety, and induced anhedonia in the control animals (*Figure 8*). The effect of prazosin and SCH23390 on control animals implies that a basal physiological level of norepinephrine and dopamine receptor activities is required for the mediation of those behaviors, while results from prazosin and SCH23390 treatment on mutant animals demonstrate that increases in norepinephrine and dopamine contribute to the mania-like behaviors of *Th-Cre;Erbb4loxp/loxp* mice.

## Discussion

We show that disruption of ErbB4 in LC-NE neurons causes NMDA receptor-mediated hyperactive spontaneous firing of LC-NE neurons and elevates CSF norepinephrine and dopamine concentrations, which induce mania-like behaviors that could be rescued by lithium or noradrenergic and dopaminergic receptor antagonists. This is the first study to demonstrate the function of ErbB4 in the regulation of behavior and mood by LC-NE neurons and of catecholamine dyshomeostasis in the pathogenesis of mania-associated disorders such as BPD.

BPD is a severe psychiatric disorder with a long-term global disease burden; however, its pathogenic mechanisms remain unknown (*Harrison et al., 2018*). Despite the increase of norepinephrine in BPD patient brains in mania episodes observed since early in the twentieth century (*Manji et al., 2003*; *Post et al., 1973*; *Post et al., 1978*), a clear description of a causal role played by norepinephrine in the pathophysiology of BPD is lacking. Here, for the first time, we demonstrates direct causality between catecholamine dyshomeostasis and mania behavior, as well as the important role of ErbB4 in BPD pathogenesis. Conditional ErbB4 deletion in LC-NE neurons increased the concentration of both norepinephrine and dopamine in the CSF, which is consistent with clinical observations of BPD patients. By specifically blocking the function of norepinephrine or dopamine, we restored the mania-like behaviors of *Th-Cre;Erbb4loxp/loxp* mice, providing strong evidence that elevated norepinephrine directly contributes to BPD pathogenesis. These findings will facilitate a better understanding of the pathophysiology of diseases associated with mania beyond BPD. The increased dopamine level may be attributable to the co-release of dopamine in NE neurons and the regulation of dopamine by NE neuronal terminals (*Carboni et al., 1990*; *Devoto et al., 2005*; *Pozzi et al., 1994*; *Yamamoto and Novotney, 1998*).

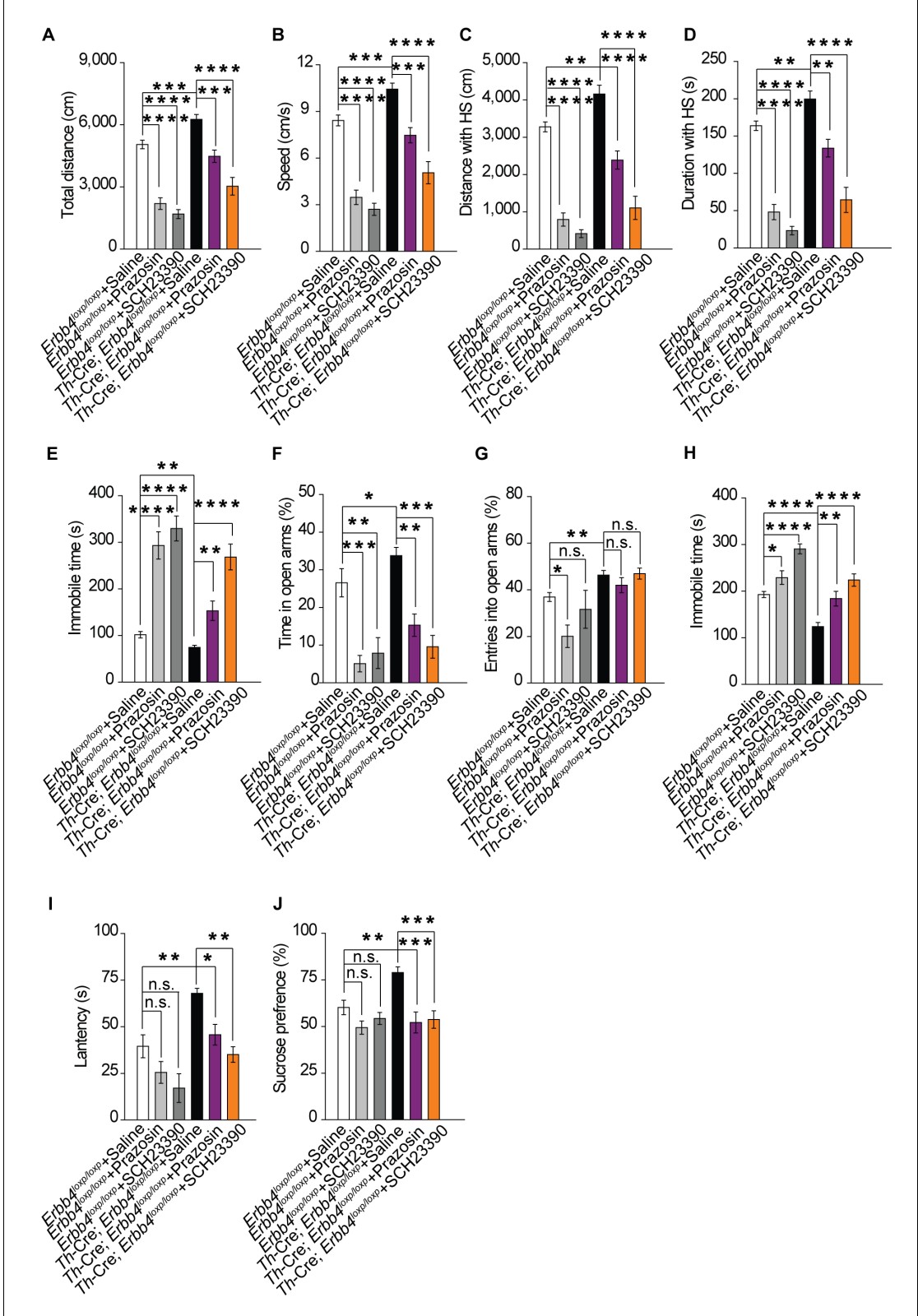

**Figure 8.** Increase in both norepinephrine and dopamine contribute to mania-like behaviors. (**A**), (**B**) Locomotor activity (**A**) and speed (**B**) of *Th-Cre; Erbb4^{loxp/loxp}* mice treated with saline (sal), prazosin, or SCH23390 in open field test. n = 17 (*Erbb4^{loxp/loxp}* + sal); n = 12 (*Erbb4^{loxp/loxp}* + prazosin); n = 11 (*Erbb4^{loxp/loxp}* + SCH23390); n = 11 (*Th-Cre;Erbb4^{loxp/loxp}* + sal); n = 10 (*Th-Cre;Erbb4^{loxp/loxp}* + prazosin); n = 9 (*Th-Cre;Erbb4^{loxp/loxp}* + SCH23390). (**C**), (**D**) Distance (**C**) and duration (**D**) at HS in open field test. (**E**) Immobility time in open field test. (**F**), (**G**) Time (**F**) and entries (**G**) in

*Figure 8 continued on next page*

**Figure 8 continued**

open arms in EPM test. n = 11 (*Erbb4^loxp/loxp* + sal); n = 12 (*Erbb4^loxp/loxp* + prazosin); n = 11 (*Erbb4^loxp/loxp* + SCH23390); n = 14 (*Th-Cre;Erbb4^loxp/loxp* + sal); n = 11 (*Th-Cre;Erbb4^loxp/loxp* + prazosin); n = 8 (*Th-Cre;Erbb4^loxp/loxp* + SCH23390). (H), (I) Immobility time (H) and latency to first surrender (I) in forced swim tests. n = 12 (*Erbb4^loxp/loxp* + sal); n = 10 (*Erbb4^loxp/loxp* + prazosin); n = 10 (*Erbb4^loxp/loxp* + SCH23390); n = 16 (*Th-Cre;Erbb4^loxp/loxp* + sal); n = 12 (*Th-Cre;Erbb4^loxp/loxp* + prazosin); n = 12 (*Th-Cre;Erbb4^loxp/loxp* + SCH23390). (J) Sucrose preference of *Th-Cre;Erbb4^loxp/loxp* mice after prazosin or SCH23390 treatment. n = 11 (*Erbb4^loxp/loxp* + sal); n = 9 (*Erbb4^loxp/loxp* + prazosin); n = 10 (*Erbb4^loxp/loxp* + SCH23390); n = 11 (*Th-Cre; Erbb4^loxp/loxp* + sal); n = 12 (*Th-Cre;Erbb4^loxp/loxp* + prazosin); n = 12 (*Th-Cre;Erbb4^loxp/loxp* + SCH23390). One-way ANOVA and Tukey's multiple comparison test. Data are expressed as means ± s.e.m. *p<0.05, **p<0.01, ***p<0.001, ****p<0.0001. n.s., not significant.

DOI: https://doi.org/10.7554/eLife.39907.030

The following source data is available for figure 8:

**Source data 1.** Statistical reporting of *Figure 8*.

DOI: https://doi.org/10.7554/eLife.39907.031

The mechanism underlying lithium treatment for BPD is complicated and unresolved (*Jope, 1999*; *Schloesser et al., 2012*). Here we showed that lithium rescued the abnormal spontaneous firing activity and NR2B expression of LC-NE neurons and alleviated mania-like behaviors in *Th-Cre;Erb-b4^loxp/loxp* mice (*Figure 7*). These observations suggest that LC-NE neurons may be a target of lithium and thus provide a possible mechanism for lithium treatment of BPD. In contrast to the manifestation in mutant mice, lithium treatment led to increased movement and decreased immobile time in the forced swim test in control mice (*Figure 7*), suggesting different functions of lithium in physiological and pathological statuses.

The LC is a nucleus consisting of most of the NE neurons in the brain, and its impairment has been implicated in many severe neurodegenerative diseases and affective disorders (*Benarroch, 2009*; *Berridge and Waterhouse, 2003*; *Aston-Jones and Cohen, 2005*; *Mather and Harley, 2016*; *Pamphlett, 2014*; *Ross et al., 2015*). Though norepinephrine has been linked with BPD, direct evidence of how the LC functions in the pathogenesis of BPD is still unclear (*Bernard et al., 2011*; *Bielau et al., 2012*; *Kato, 2008*). Here, we provide direct evidence demonstrating the crucial role of the LC in BPD pathogenesis.

Work by *D'Andrea et al. (2015)* showed that LC neuronal dysfunction by CREB signaling hyperactivity cause ADHD-like behavior in PI3Kγ conventional knockout (PI3Kγ KO) mice. Since ADHD and mania animal models have overlapped behavioral phenotypes (*Beyer and Freund, 2017*; *Itohara et al., 2015*), we treated ErbB4-deficient mice with methylphenidate (MHP), a clinical ADHD medication which alleviates hyperactivity of ADHD animal models but aggravates hyperactivity of mania models (*D'Andrea et al., 2015*; *Souza et al., 2016*; *Sumitomo et al., 2018*). Our results showed that MHP aggravated hyperactivity of ErbB4-deficient (*Erbb4^loxp/loxp* + AAV-Cre-GFP) mice (*Figure 5—figure supplement 3*). In addition, the CREB signaling in the LC was unaltered in our LC ErbB4-deficient mice (*Figure 5—figure supplement 4*). Thus, these results suggest that our LC ErbB4-deficient mice should be a mania model.

Using ELISA kit, D'Andrea et al. showed that PI3Kγ KO mice showed increased norepinephrine and decreased dopamine levels in prefrontal cortex and striatum (*D'Andrea et al., 2015*), while both norepinephrine and dopamine were elevated in the CSF of LC ErbB4-deficient mice in our study. The possible explanation might be that the specimens and measurement methods of their and our studies were different and that the decrease of dopamine in PI3Kγ KO mice may be caused by the regulation of other PI3Kγ KO non-catecholaminergic neurons. Meanwhile, the dysregulation of different molecular players (e.g. ErbB4, PI3Kγ) in LC-NE neurons may lead to different cellular states that causes diverse psychiatric–like behaviors.

*Erbb4* is a genetic susceptibility gene for schizophrenia (*Mei and Xiong, 2008*; *Pitcher et al., 2011*), with many studies reporting the crucial role of ErbB4 in the pathogenesis of schizophrenia (*Chong et al., 2008*; *Del Pino et al., 2013*; *Hahn et al., 2006*; *Shamir et al., 2012*). While coding variants of *Erbb4* have also been genetically associated with BPD (*Chen et al., 2012*; *Bipolar Genome Study et al., 2011*), direct evidence remains limited. We reveal that functional deficiency of ErbB4 in LC-NE neurons facilitates the paroxysm of mania-like behaviors and increases spontaneous firing of LC-NE neurons (*Figure 2A–D*). In contrast, conditional ErbB4 deletion in parvalbumin-positive GABAergic neurons in the frontal cortex decreases the excitability of these

neurons via KV$_{1.1}$ (*Kx et al., 2012*). These lines of evidence suggest that ErbB4 may function differently in different neurons.

The NMDA receptor, especially its subunit isoform NR2B, is regulated by ErbB4 in the hippocampus and prefrontal cortex (*Bjarnadottir et al., 2007*; *Pitcher et al., 2011*). Consistent with previous studies on the influence of ErbB4 on NR2B, we observed NR2B overexpression and enhanced NMDAR function in LC tissue of ErbB4-deficient mice (*Figure 3*). However, how ErbB4 deletion increases NR2B protein expression and how NR2B overexpression participates in the strengthening of NMDA receptor function in the hyperexcitability of ErbB4-deficient NE neurons requires further investigation.

Past studies have reported ErbB4 mRNA to be highly expressed in the LC area (*Gerecke et al., 2001*). We confirmed ErbB4 protein expression in LC-NE neurons (*Figure 1F, G*). In addition, we functionally validated the presence of ErbB4 in LC-NE neurons by showing increased spontaneous firing and enhanced NMDAR function upon ErbB4 deletion. However, a previous report failed to detect the expression of Cre/Tomato in LC neurons in *ErbB4::CreERT2;Rosa::LSL-tdTomato* mice (*Bean et al., 2014*). This discrepancy may arise from the different experimental methods adopted by the different research groups.

Earlier research has shown that Cre is abundantly expressed in *Th-Cre* mouse lines in the NE and dopaminergic neurons of the LC and midbrain, respectively (*Lindeberg et al., 2004*; *Savitt et al., 2005*). Recent research used the *Th-Cre* line (*Gong et al., 2007*) to drive ErbB4 selective deletion, with gene loss mainly observed in dopaminergic neurons in the midbrain (*Gong et al., 2007*; *Skirzewski et al., 2017*). In contrast, very low Cre expression was detected in the midbrain dopaminergic neurons in our *Th-Cre* mice (*Gelman et al., 2003*) compared with the abundant Cre expression observed in the LC-NE neurons. Though mice of the same genotype (both *Th-Cre;Erbb4$^{loxp/loxp}$*) were used, the varied Cre expression in the distinct *Th-Cre* lines in our research and that of *Skirzewski et al. (2017)* yielded different findings on ErbB4 function in different neuronal types. ErbB4 deletion in dopaminergic neurons in the midbrain led to deficits in spatial/working memory but had no influence on locomotion or anxiety (*Skirzewski et al., 2017*). In comparison, our mutant mice with ErbB4 deletion in LC-NE neurons presented significant hyperactivity and reduced anxiety (*Figure 4*). The reason underlying the discrepancy between different *Th-Cre* lines is currently unknown (*Lammel et al., 2015*). One probable explanation may be that *Cre* is inserted into different chromosomal loci and the surrounding genetic or epigenetic elements may modify the spatial and temporal regulation of *Cre* gene expression. Nevertheless, discrepancy of behaviors presented by ErbB4 mutants used by Skirzewski et al. and by us is an additional line of evidence supporting that ErbB4 deletion in LC-NE neurons, instead of off-target ErbB4 deletion in catecholaminergic neurons in other brain regions, is the primary cause for abnormalities of our *Th-Cre;Erbb4$^{loxp/loxp}$* mice.

Together, our findings demonstrate the importance of ErbB4 in LC-NE neurons in behavior and mood regulation and reveal the participation of catecholamine homeostasis modulated by ErbB4 in the pathogenesis of mania-associated disorders. Future studies aimed at identifying ErbB4 downstream signals in LC-NE neurons may provide new insights into therapies for mania-associated disorders.

## Materials and methods

### Generation and maintenance of mice

Four mouse lines were used. We first crossed *Th-Cre* mice (kindly provided by Yuqiang Ding, Tongji University School of Medicine, Shanghai, China), which have been described previously (*Gelman et al., 2003*), with *Erbb4$^{loxp/loxp}$* mice (Mutant Mouse Regional Resource Center from North America), generating a *Th-Cre;Erbb4$^{loxp/loxp}$* mouse line in which ErbB4 was mainly deleted in the noradrenergic neurons (NE neurons). For Immunohistochemical analysis and electrophysiological recordings, we then crossed *Th-Cre;Erbb4$^{loxp/loxp}$* mice with Ai9 mice or Ai3 mice, which are used as a Cre reporter strain (purchased from Jackson Laboratory). *Th-Cre* mice were in C57BL/6N genetic background and *Erbb4$^{loxp/loxp}$* mice were in the C57BL/6 genetic background (No substrain information was available from Mutant Mouse Regional Resource Center from North America, Stock Number: 010439). The control mice used in behavior experiments were littermates to the *Th-Cre;Erbb4$^{loxp/loxp}$* mice. Additionally, we have compared the behavioral performance between *Th-Cre*

mice and *Erbb4^{loxp/loxp}* mice, which were littermates to the *Th-Cre; Erbb4^{loxp/loxp}* mice, and there was no significant difference between these two mouse lines (data not shown). Only male mice (8 – 12 weeks old) with normal appearance and weight were used in experiments and were divided into different groups randomly. All mice were housed under a 12 hr light/dark cycle (lights were on from 7:00 am-7:00 pm everyday) and had access to food and water *ad libitum*.

## Immunohistochemical analysis

Mice were anesthetized with 10% chloral hydrate and perfused with ice-cold saline followed by para-formaldehyde (PFA) (4%) in phosphate-buffered saline (PBS). Brains were removed and fixed in the same 4% PFA solution at 4°C overnight and transferred to 30% sucrose in PBS for 2 d. Frozen brains were sectioned at 30 µm with a sliding microtome (Leica CM3050 S, Leica biosystems) in the coronal plane. Slices were immersed in PBS with 0.02% sodium azide and stored at 4°C until further use. After incubation in blocking buffer containing 5% goat serum and 3% bovine serum albumin (BSA) in PBST (0.5% Triton X-100 in PBS) for 1 hr at room temperature, slices were incubated with primary antibodies (rabbit tyrosine hydroxylase (TH)-specific antibody (1:700, Abcam), mouse ErbB4-specific antibody (1:300, Abcam)) in blocking buffer at 4°C overnight. The slices were washed three times in PBST and incubated with Alexa Fluor 488- or Alexa Fluor 543-conjugated secondary antibodies at 25°C for 1 hr. All slices were counterstained with DAPI during final incubation. Fluorescent image acquisition was performed with an Olympus FluoView FV1000 confocal microscope using a 20 × objective lens and analyzed using ImageJ software.

## Western blot analysis

In western blotting experiments, our controls were *ErbB4^{loxP/loxP}* mice or *ErbB4^{loxP/loxP}* + AAV-GFP mice. Brain tissues from control and *Th-Cre;Erbb4^{loxp/loxp}* mice or *Erbb4^{loxp/loxp}* + AAV-Cre-GFP mice were homogenized in RIPA lysis buffer containing 50 mM Tris (pH 7.4), 150 mM NaCl, 1% Triton X-100, 1% sodium deoxycholate, 0.1% SDS, 1 mM PMSF, and phosphatase inhibitor cocktail (Cell Signaling Technology). Protein samples were loaded on 10% acrylamide SDS-PAGE gels and then transferred to nitrocellulose membranes. After incubation with 4% BSA for 1 hr at 25°C, membranes were incubated with primary antibodies at 4°C overnight (sheep TH-specific antibody, 1:2,000, Millipore; rabbit TH-Ser40-specific antibody, 1:1,000, Millipore; rabbit ErbB4-specific antibody, 1:2,000, Abcam; rabbit norepinephrine transporter (NET)-specific antibody, 1:300, Millipore; rabbit dopamine beta-hydroxylase (DBH)-specific antibody, 1:300, Abcam; rabbit GAPDH-specific antibody, 1:5,000, Cell Signaling Technology; mouse catechol-o-methyltransferase (COMT)-specific antibody, 1:5,000, BD Biosciences; and rabbit actin-specific antibody, 1:2,000, Cell Signaling Technology). The membranes were washed three times and then incubated for 1 hr with horseradish peroxidase-conjugated secondary antibodies in 4% BSA at 25°C. Immunoreactive bands were visualized clearly by X-ray film exposure (ECL kit, Thermo Scientific) and analyzed using NIH ImageJ software. Each experiment was repeated at least three times.

## Surgery and microdialysis

Mice were deeply anesthetized with isoflurane (0.15% in oxygen gas) and mounted on a stereotaxic frame (RWD Life Science). A stainless steel guide cannula with a dummy probe was implanted into the lateral ventricle (anteroposterior (AP) = −0.6 mm; mediolateral (ML) =± 1.2 mm; dorsoventral (DV) = −2.0 mm). After 7 d of recovery, the dummy probe was replaced with a microdialysis probe (membrane length: 4 mm, molecular weight cut-off: 18,000 Da, outer diameter: 0.2 mm). For balance, artificial cerebrospinal fluid (ACSF), which contained (in mM) 125 NaCl, 2.5 KCl, 2 CaCl$_2$, 1 MgCl$_2$, 1.25 NaH$_2$PO$_4$, 25 NaHCO$_3$, and 11 D-glucose, was perfused continuously by syringe pump at a speed of 2 µl min$^{-1}$ for 2 hr before sample collection. Samples (60 µl each) were automatically collected from each mouse for 2 hr and analyzed by high-performance liquid chromatography (HPLC) with an electrochemical detector (5014b, ESA, USA). The concentrations of norepinephrine and dopamine were detected by HPLC (Coulochem III, ESA, USA) using a C18 column (MD150 3 mm × 150 mm, 5 µm, ESA, USA).

## Slice preparation

Mice were deeply anesthetized and decapitated. The brain was quickly removed and immersed in ice-cold high-sucrose ACSF bubbled with 95% $O_2$/5% $CO_2$ to maintain a pH of 7.4. High-sucrose ACSF contained the following (in mM): 200 sucrose, 3 KCl, 2 $CaCl_2$, 2 $MgCl_2$, 1.25 $NaH_2PO_4$, 26 $NaHCO_3$, and 10 D-glucose. Coronal slices (250 µm) were prepared with a vibratome (Leica, VT 1000S, Germany), allowed to rest for 1 hr at 34°C in oxygenated ACSF, and then maintained at 25°C before transfer to the recording chamber.

## Electrophysiology

Acute slices from adult control mice (*Ai9;Th-Cre* mice or *Erbb4^{loxp/loxp}* mice injected with AAV-GFP virus), or *Ai9;Th-Cre;Erbb4^{loxp/loxp}* mice, or *Erbb4^{loxp/loxp}* mice injected with AAV-Cre-GFP virus were transferred to a recording chamber and fully submerged in ACSF at 25°C, which was continuously perfused (2 ml/min) with oxygen. Fluorescent neurons were visually identified under an upright microscope (Nikon, Eclipse FN1) equipped with an infrared-sensitive CCD camera. Electrophysiological recordings were performed in cell-attached mode for spontaneous firing recording and in whole-cell mode for detection of NMDAR current, sEPSC, sIPSC, and intrinsic membrane properties by MultiClamp 700B Amplifier equipped with Digidata 1440A analog-to-digital converter. For intrinsic membrane properties and spontaneous firing recordings, microelectrodes (3 – 5 MΩ) were filled with a solution containing 130 mM potassium gluconate, 20 mM KCl, 10 mM HEPES buffer, 2 mM $MgCl_2$·6 $H_2O$, 4 mM Mg-ATP, 0.3 mM Na-GTP, and 10 mM EGTA; the pH was adjusted to 7.25 with 10 M KOH. 3 mins for stabilization. AP-V (50 µM, Tocris Bioscience), DNQX (30 µM, Tocris Bioscience) and picrotoxin (50 mM) were present in the bath solution for intrinsic membrane properties recordings. Spontaneous firing was recorded for at least 4 min for each neuron. For NMDAR current and sEPSC recording, microelectrodes (3 – 5 MΩ) were filled with a solution containing 140 mM Cs-methanesulfonate, 5 mM NaCl, 1 mM $MgCl_2$.6$H_2O$, 10 mM HEPES, 0.2 mM EGTA, 2 mM MgATP, 0.5 mM NaGTP, 0.5 mM spermine, and 5 mM QX314 Chloride; the pH was adjusted to 7.25 with 10 M CsOH. To isolate sEPSC, picrotoxin (50 mM) was present in the bath solution. AMPA-sEPSC were recorded at −60 mV with DL-2-amino-5-phosphonopentanoic acid (AP-V; 50 µM, Tocris Bioscience, to block NMDA receptors) in the bath solution. NMDAR-EPSC were induced using bipolar electrodes and recorded at +40 mV with 6, 7-dinitroquinoxaline-2, 3 (1H, 4H)-dione (DNQX; 30 µM, Tocris Bioscience, to block AMPA receptors) in the bath solution. For sIPSC recording, microelectrodes (3 – 5 MΩ) were filled with a solution containing 120 mM CsCl, 20 mM Cs-methanesulfonate, 5 mM NaCl, 1 mM $MgCl_2$.6$H_2O$, 10 mM HEPES, 0.2 mM EGTA, 2 mM MgATP, 0.5 mM NaGTP, 0.5 mM spermine, and 5 mM QX314 Chloride; the pH was adjusted to 7.25 with 10 M CsOH. To isolate sIPSC, AP-V (50 µM, Tocris Bioscience) and DNQX (30 µM, Tocris Bioscience) were present in the bath solution. Electrophysiological recordings were performed at the same time of day for control (*Erbb4^{loxp/loxp}* mice) and *Th-Cre;Erbb4^{loxp/loxp}* mice from 13:00 to 17:00 on the each experimental day. All analyses were performed using Clampfit 10.2 (Axon Instruments/Molecular Devices), Minianalysis, and Matlab software.

## Virus vectors and stereotactic injection

AAV-GFP and AAV-Cre-GFP carrying human synaptophysin promotor for gene expression were purchased from Shanghai SBO Medical Biotechnology Company, Shanghai. For viral injection, 1-month-old *Erbb4^{loxp/loxp}* mice were anesthetized with chloral hydrate (400 mg/kg of body weight) by intra-peritoneal (i.p.) injection and placed in a stereotactic frame, with their skulls then exposed by scalpel incision. Glass microelectrodes were used to bilaterally inject 0.15 µl of purified and concentrated AAV (~$10^{11}$ infections units per ml) into the locus coeruleus (LC) (coordinates from bregma: anterior-posterior, 5.25 mm; lateral-medial, 1.00 mm; dorsal-ventral, –4.5 mm) at 100 nl min$^{-1}$. The injection microelectrode was slowly withdrawn 2 min after virus infusion.

## Behavioral assays

All experiments were performed in quiet rooms (<40 dB) equipped with dumboard between 13:00 and 16:00 and analyzed in a double-blind fashion. According to the principal of beginning with the test with minimized stress stimulation, each mouse was subjected to behavioral tests in the following order: open field test, elevated plus maze test, 6 min forced swim test, and lastly, the most time-

consuming sucrose preference test. For the first three tests, mice were rested in their home cage for 1 – 2 days between two behavior tests. For the last one, mice were rested in their home cage for 3 days before the test.

### Open field test

Open field tests were performed in an open field chamber (50 cm × 50 cm) equipped with infrared sensors (CCTV lens) in a room with dim light (18 lux). Mice could freely explore the novel environment for 10 min, and their movements were traced and analyzed simultaneously using viewpoint application manager software (VideoTrack 3.10). Total distance, speed, and immobility time were analyzed. The open field chamber was cleaned with 70% ethanol and wiped with paper towels between tests.

### Elevated plus maze test

Elevated plus maze (EPM) tests were performed in a dimly lit room (8 lux). The maze was elevated 70 cm above the floor and consisted of two closed arms (5 × 30 cm) surrounded by 15-cm-high plastic walls and two open arms (5 × 30 cm). For testing, a mouse was placed in the center (5 × 5 cm) of the maze and allowed to explore for 5 min. Mouse movements were recorded and analyzed using Mobile Datum recording and analysis software. The amount of time spent in and number of entries into the open arms and closed arms were measured. The maze was cleaned with 70% ethanol and wiped with paper towels after each test.

### Forced swim test

Forced swim tests were performed in a room with normal light (54 lux). Mice were placed in a transparent plastic cylinder (diameter: 12 cm; height: 30 cm) containing 20 – 24°C water at 15 cm depth. During the 6 min test period, the mice were monitored using a video camera (Mobile Datum) from the side. Total time spent, immobility time, and latency to first immobility were analyzed by an observer off-line, who was blinded to the experimental treatments. After the 6 min test period, the cylinders were cleaned with 70% ethanol and wiped with paper towels. The water in the cylinder was changed for each new mouse.

### Sucrose preference test

Mice were single-housed for 1 week with a normal drinking water bottle. The bottle was then replaced with two identical bottles (bottle 'A' and bottle 'B') filled with drinking water for 2 d (W/W). The positions of bottle A and bottle B were switched daily to avoid place preference. Bottle A and bottle B were then filled with drinking water alone and drinking water with 2% sucrose, respectively, for 2 d (W/S) and switched after 24 hr. The consumption of the solutions in bottle A and bottle B were measured by weighing, and the preference for sucrose was calculated as the ratio of consumption of the sucrose solution to that of both the water and sucrose solutions during the 2 d of testing.

### Prepulse inhibition

During the PPI test, mice were subjected to 20 startle trials (120 dB, 20 ms), 10 pre-pulse/startle trials (pre-pulse duration, 20 ms; intensities, 75 dB, 80 dB, and 85 dB; interstimulus intervals, 100 ms; and 20 ms 120 dB startle stimulus), 15 pre-pulse trials (5 for 75 dB, 80 dB, and 85 dB each), and five background noise trials (65 dB), for a total of 70 trials. Different trial types were presented pseudorandomly. No two consecutive trials were identical except for five consecutive startle trials at the beginning and end of each session, which were not used for PPI analysis. Mouse movement was measured during 65 ms after startle stimulus onset (sampling frequency 1 kHz). PPI (%) was calculated according to the formula: (100 − (startle amplitude on pre-pulse-startle trials /startle amplitude on startle pulse alone trials) × 100).

### Drug treatment

Mice were treated for 10 d with lithium chloride (600 mg L$^{-1}$) in drinking water and were then subjected to behavioral tests or sacrificed for Western blotting or patch clamp experiments (*Dehpour et al., 2002*; *Roybal et al., 2007*). Methylphenidate (MHP) (10 mg/kg), prazosin (1 mg/kg), or SCH23390 (0.125 mg/kg) were injected (i.p.) 30 min before behavioral experiments.

## Quantification and statistical analysis

The samples were randomly assigned to each group and restricted randomization was applied. The investigator was blinded to group allocation and when assessing outcome in the all behavioral tests and immunocytochemistry tests. For the electrophysiology experiments, the investigator was blinded when assessing outcome. For quantification, values from three independent experiments with at least three biological replicates were used. For behavioral assays, all population values appeared normally distributed, and variance was similar between groups. Sample size was calculated according to the preliminary experimental results and the formula: $N = [(Z_{\alpha/2} + Z_{\beta})\sigma / \delta]^2 (Q1^{-1} + Q2^{-1})$, where $\alpha = 0.05$ significance level, $\beta = 0.2$, power = $1-\beta$, $\delta$ is the difference between means of two samples, and Q is the sample fraction. All data are presented as means $\pm$ s.e.m. and were analyzed using two-tailed Student's t-test, one-way analysis of variance (ANOVA), two-way ANOVA, or two-way repeated-measures ANOVA. The Kolmogorov–Smirnov test (K–S test) was used to compare the interspike interval distributions, as specified in each figure legend and source data of *Figure 1–7*. Grubbs' test are used to detect an outlier. All data were analyzed using Origin8.0 (OriginLab). Data were exported into Illustrator CS5 (Adobe Systems) for preparation of figures.

## Acknowledgements

We thank Nick Spitzer (University of California at San Diego) for manuscript suggestions. We are grateful to YQ Ding (Tongji University) and XQ Chen (Institute of Neuroscience, Zhejiang University) for providing mouse lines or experimental facilities. We express our thanks to L Wang (Zhejiang University) for technical assistance. We are grateful to the Core Facilities of Zhejiang University Institute of Neuroscience for technical assistance of behavioral experiments. This work was also supported by the Non-profit Central Research Institute Fund of Chinese Academy of Medical Sciences (2017PT31038 and 2018PT31041).

## Additional information

### Funding

| Funder | Grant reference number | Author |
| --- | --- | --- |
| National Key R&D Program of China | 2016YFA0501003 | Xiao-Ming Li |
| National Natural Science Foundation of China | 31700904 | Shu-Xia Cao |
| Zhejiang Provincial Natural Science Foundation of China | LY17C090004 | Hong Lian |

The funders had no role in study design, data collection and interpretation, or the decision to submit the work for publication.

### Author contributions

Shu-Xia Cao, Data curation, Formal analysis, Funding acquisition, Investigation, Methodology, Writing—original draft, Project administration, Writing—review and editing; Ying Zhang, Data curation, Investigation, Methodology; Xing-Yue Hu, Tian-Ming Gao, Investigation, Methodology; Bin Hong, Peng Sun, Hai-Yang He, Hong-Yan Geng, Investigation; Ai-Min Bao, Methodology; Shu-Min Duan, Jian-Ming Yang, Methodology, Writing—review and editing; Hong Lian, Funding acquisition, Investigation, Methodology, Project administration, Writing—review and editing; Xiao-Ming Li, Resources, Supervision, Funding acquisition, Validation, Project administration, Writing—review and editing

### Author ORCIDs

Shu-Xia Cao (iD) http://orcid.org/0000-0002-2096-5518
Hong Lian (iD) http://orcid.org/0000-0003-3835-6590
Xiao-Ming Li (iD) http://orcid.org/0000-0002-8617-1702

## Ethics

Animal experimentation: This study was performed in strict accordance with the recommendations in the Guide for the Care and Use of Laboratory Animals of the Zhejiang University. The care and use of the mice in this work were reviewed and approved by the Animal Advisory Committee at Zhejiang University (ZJU201553001). Every effort was made to minimize suffering.

## Decision letter and Author response

Decision letter https://doi.org/10.7554/eLife.39907.034
Author response https://doi.org/10.7554/eLife.39907.035

# Additional files

## Supplementary files

• Transparent reporting form
DOI: https://doi.org/10.7554/eLife.39907.032

## Data availability

All data generated or analysed during this study are included in the manuscript and supporting files. Source data files have been provided for Figures 1, 2, 3, 4, 5, 6, 7 and 8.

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
