## [Decision Letter]

[Editors’ note: a previous version of this study was rejected after peer review, but the authors submitted for reconsideration. The first decision letter after peer review is shown below.]

Thank you for submitting your work entitled "ErbB4 deletion in noradrenergic neurons in the locus coeruleus induces mania-like behavior via elevated catecholamines" for consideration by *eLife*. Your article has been reviewed by three peer reviewers, one of whom is a member of our Board of Reviewing Editors, and the evaluation has been overseen by a Senior Editor. The following individuals involved in review of your submission have agreed to reveal their identity: Kihoon Han (Reviewer #2).

Our decision has been reached after consultation between the reviewers. Based on these discussions and the individual reviews below, we regret to inform you that your work will not be considered further for publication in *eLife*.

Although the reviewers find the study interesting, it is our policy not to invite revision if the revision experiments are going to take significantly longer than two months. However, if you are willing to address all the review comments and resubmit the manuscript to *eLife*, we will be happy to consider your manuscript again.

Reviewer #1:

This study by Cao et al. reports the characterization of mice lacking ErbB4 lacking in LC-NE neurons. LC-NE neurons lacking ErbB4 neurons display enhanced neuronal firing associated with increased NMDAR function and NE and DA levels. Behaviorally, these mice show hyperactivity and suppressed anxiety-like behavior and depression-like behavior. These behaviors are recapitulated by local ErbB4 deletion mediated by Cre virus injection and rescued by the treatment of lithium, NE antagonist, and DA antagonist. Lithium also rescues tyrosine hydroxylase activity and spontaneous firing in addition to behaviors.

1) In Figure 3, the authors suggest that LC-NE neurons show increased spontaneous firing due to increased NMDAR activity. However, this conclusion should be supported by additional experiments. The authors did not show enhanced NMDAR currents directly. In addition, AP5 may be able to suppress the enhanced spontaneous firing, but the enhanced neuronal firing could be induced by other mechanisms such as alterations in the balance between excitatory synapses and inhibitory synapses and also the intrinsic excitability of the neurons. Therefore, the authors should put more effort to determine potential changes in NMDAR function, spontaneous/evoked excitatory/inhibitory transmission, or intrinsic excitability.

Reviewer #2:

In this study, Dr. Xiao-Ming Li and colleagues generated locus coerulesu (LC)-specific conditional knock-out mice for ErbB4 (using either *Th-Cre* mice or AAV-Cre), and showed that the mice exhibited mania-like behaviors responsive to lithium. As a cellular mechanism, LC neurons of the KO mice had increased spontaneous firing possibly due to NMDA receptor hyperfunction, which elevated the concentrations of norepinephrine and dopamine in the CSF. Overall, the manuscript is well-written and the results provide new insights into the role of ErbB4 in LC in pathophysiology of mania.

1) In 2015 D'Andrea et al. reported that PI3Kγ KO mice displayed attention-deficit/hyperactivity disorder (ADHD)-like behaviors by increasing CREB signaling in the LC (EMBO molecular medicine (2015): e201404697.). Moreover, it was shown that expression of constitutively active form of CREB increased firing rate of the LC neurons (Journal of Neuroscience 26.17 (2006): 4624-4629.). Considering these previous reports, it is important to confirm that the LC-specific ErbB4 KO mice are true a manic model, but not an ADHD model. The behavioral effects of methylphenidate or amphetamine can be tested, which usually shows paradoxical calming effect in ADHD models but aggravating effect in manic models of their hyperactivity.

2) Lithium treatment rescued mania-like behaviors and neuronal firing rate of the KO mice. It is necessary to show whether lithium treatment also normalized the expression level of NR2B. If the NR2B level was not affected by lithium, decreased tyrosine phosphorylation of NR2B may be involved (Journal of neurochemistry 80.4 (2002): 589-597.).

Reviewer #3:

The study entitled "ErbB4 deletion in noradrenergic neurons in the locus coeruleus induces mania-like behavior via elevated catecholamines" by Shu-Xia Cao, Ying Zhang and colleagues uses genetic and viral vector approaches to investigate whether ErbB4 regulates noradrenergic neuron function in the locus coeruleus (LC) and mania-like behaviors. The majority of the experiments looked at LC function and mania-like behaviors in *Th-Cre;Erbb4^loxp/loxp^* mice, which limits the interpretation of these findings since it cannot be ruled out that the effects could be mediated by loss of ErbB4 in other brain regions. Importantly the authors showed that viral-mediated deletion of ErbB4 in the LC recapitulated the mania-like behaviors, but no experiments were done to show that LC-specific deletion of ErbB4 recapitulates any of the effects on LC function. Furthermore, it was stated that 77% of the Cre/GFP+ neurons were TH^+^, the authors should clarify if there was off-target ErbB4 deletion. Interestingly, lithium reversed the mania-like behaviors of *Th-Cre;Erbb4^loxp/loxp^* mice. However, enthusiasm was dampened for the norepinephrine and dopamine antagonist studies since the authors did not include control mice treated with the antagonists. This work is important and intriguing, but the conclusions would be better supported after some revisions and clarifications.

The authors could quantify the immunofluorescent images for Figure 1F. At a minimum, it would be better for Figure 1F to show an image of ErbB4 in the entire LC of *Th-Cre;Erbb4^loxp/loxp^* mice as opposed to a high magnification image of a few cells.

Please clarify whether electrophysiology experiments were done at the same time of day for control and *Th-Cre;Erbb4^loxp/loxp^* mice.

The authors should clarify what the control mice are in the electrophysiology and western blot studies. Differences in genetic background or Cre-recombinase itself could affect the reported measures.

Please report the number of mice used per group for the behavior experiments.

The authors should be careful in saying that ErbB4 was predominantly deleted from noradrenergic neurons. The authors should clarify that ErbB4 was determined to be predominantly deleted from noradrenergic neurons in the locus coeruleus in the midbrain (Figure 1). The authors did not look at ErbB4 expression in other brain regions with catecholaminergic neurons.

Please clarify how% open arm entries and% open arm time were calculated. It appears that the values were normalized to the control mice which seems unnecessary. Values were not normalized for other experiments. Typically,% open arm entries refers to the number of open arm entries relative to total arm entries expressed as a percentage, which controls for differences in overall activity.

If the authors have the data it would be interesting to know whether *Th-Cre;Erbb4^loxp/loxp^* mice display differences in spatial d and center entries/time in the open field.

For the graphs showing inter-spike intervals, the authors may consider labeling the y-axis as "cumulative probability" as opposed to "cumulative possibility".

The authors should describe the order of behavior testing, how many behavior tests the mice were subjected to, and the rest periods in between behavior testing.

The authors give the time of day that the behavior tests were ran, but do not describe the 12 h light/dark schedule of the mice. Please include the lights on and off housing schedule.

The authors should clarify what promoter was used for the control AAV-GFP virus.

[Editors’ note: minor issues and corrections have not been included, so there is not an accompanying Author response.]

Thank you for resubmitting your work entitled "ErbB4 deletion in noradrenergic neurons in the locus coeruleus induces mania-like behavior via elevated catecholamines" for further consideration at *eLife*. Your revised article has been reviewed by three peer reviewers, one of whom is a member of our Board of Reviewing Editors, and the evaluation has been overseen by a Senior Editor.

The manuscript has been improved but there are some remaining issues that need to be addressed before acceptance, as outlined below:

1) Please provide more detailed information about the mouse substrain (C57BL/6J or C57BL/6N). This is important because those substrains have distinct behavioral phenotypes (Kumar et al., Science. 2013 Dec 20; 342(6165): 1508-1512.).

2) Please provide light (in lux) and sound (in dB) intensities during the behavioral tests. Did the authors use white noise as background sound?

---

## [Author Response]

[Editors’ note: the author responses to the first round of peer review follow.]

We appreciate the time and effort the editors and three reviewers have put to find our study interesting and give us such critical and constructive comments to improve the manuscript. We have performed all of the experiments the reviewers required and fully addressed all concerns. We believe that the quality of this manuscript has been really improved.

Reviewer #1:[…] 1) In Figure 3, the authors suggest that LC-NE neurons show increased spontaneous firing due to increased NMDAR activity. However, this conclusion should be supported by additional experiments. The authors did not show enhanced NMDAR currents directly. In addition, AP5 may be able to suppress the enhanced spontaneous firing, but the enhanced neuronal firing could be induced by other mechanisms such as alterations in the balance between excitatory synapses and inhibitory synapses and also the intrinsic excitability of the neurons. Therefore, the authors should put more effort to determine potential changes in NMDAR function, spontaneous/evoked excitatory/inhibitory transmission, or intrinsic excitability.

These are great suggestions. We performed lots of new experiments to further investigate the mechanisms underlying the enhanced neuronal firing. NMDAR current, spontaneous EPSC/IPSC (sEPSC/sIPSC), and action potential were recorded on LC-NE neurons of brain slices from the *Ai9;Th-Cre* (Control) and *Ai9;Th-Cre;ErbB4^loxp/loxp^* mice (*Th-Cre; ErbB4^loxp/loxp^*) in which Ai9 is the reporter line showing red fluorescence upon Cre expression that allows identification of noradrenergic neurons to detect ErbB4-dependent alterations in NMDAR function, excitation-inhibition balance, and intrinsic excitability of the LC-NE neurons.

Firstly, as shown in revised Figure 3C-D, the amplitude of evoked NMDAR current was significantly increased in the LC-NE neurons with ErbB4 deletion compared with the controls.

Secondly, spontaneous EPSC/IPSC (sEPSC/sIPSC) and intrinsic properties of LC-NE neurons are shown in revised Figure 3—figure supplement 1 and Figure 3—figure supplement 2, respectively. Our results indicate that loss of ErbB4 has no effect on the amplitude and frequency of either AMPAR-sEPSC or sIPSC (Figure 3—figure supplement 1). In addition, the intrinsic properties of LC-NE neurons including AP threshold, AP amplitude, AP half-width, afterhyperpolarization (AHP), rheo-based current, C_m_, R_in_, and τ were unaltered (Figure 3—figure supplement 2).

Reviewer #2:[…] 1) In 2015 D'Andrea et al. reported that PI3Kγ KO mice displayed attention-deficit/hyperactivity disorder (ADHD)-like behaviors by increasing CREB signaling in the LC (EMBO molecular medicine (2015): e201404697.). Moreover, it was shown that expression of constitutively active form of CREB increased firing rate of the LC neurons (Journal of Neuroscience 26.17 (2006): 4624-4629.). Considering these previous reports, it is important to confirm that the LC-specific ErbB4 KO mice are true a manic model, but not an ADHD model. The behavioral effects of methylphenidate or amphetamine can be tested, which usually shows paradoxical calming effect in ADHD models but aggravating effect in manic models of their hyperactivity.

We thank the reviewer for the valuable suggestion. To address reviewer's concern, we performed methylphenidate (MHP) and saline injection to viral-mediated LC-specific ErbB4 deletion mice (*ErbB4^loxp/loxp^*+AAV-Cre-GFP) and detected their behavioral responses. MHP increased locomotor activity of control mice (*ErbB4^loxp/loxp^*+AAV-GFP) and as the reviewer described, aggravated hyperactivity of ErbB4-deficient mice (revised Figure 5—figure supplement 3).

In addition, we did biochemical analysis to identify whether our ErbB4-deficient mice might be an ADHD model. Using protein lysates from the LC tissue from the control (*ErbB4^loxp/loxp^*) and *Th-Cre; ErbB4^loxp/loxp^* mice, we measured the total CREB protein level and active CREB form, phosphorylated CREB (pCREB) at Ser133, to determine the activity of the CREB signaling upon ErbB4 deletion in the LC (revised Figure 5—figure supplement 4). Neither total CREB nor CREB phosphorylation was changed in the mutant mice compared to the controls. We have discussed this in the fifth paragraph of the Discussion.

2) Lithium treatment rescued mania-like behaviors and neuronal firing rate of the KO mice. It is necessary to show whether lithium treatment also normalized the expression level of NR2B. If the NR2B level was not affected by lithium, decreased tyrosine phosphorylation of NR2B may be involved (Journal of neurochemistry 80.4 (2002): 589-597.).

This is good suggestion. We tested NR2B protein expression of lithium-treated control mice (*ErbB4^loxp/loxp^*) and ErbB4 conditional knockout mice *(Th-Cre; ErbB4^loxp/loxp^*). Similar to the rescuing effect on behaviors and firing rate, lithium normalized NR2B overexpression in the mutants to the control level (Figure 7K-L).

Reviewer #3:The study entitled "ErbB4 deletion in noradrenergic neurons in the locus coeruleus induces mania-like behavior via elevated catecholamines" by Shu-Xia Cao, Ying Zhang and colleagues uses genetic and viral vector approaches to investigate whether ErbB4 regulates noradrenergic neuron function in the locus coeruleus (LC) and mania-like behaviors. The majority of the experiments looked at LC function and mania-like behaviors in Th-Cre;Erbb4^loxp/loxp^ mice, which limits the interpretation of these findings since it cannot be ruled out that the effects could be mediated by loss of ErbB4 in other brain regions. Importantly the authors showed that viral-mediated deletion of ErbB4 in the LC recapitulated the mania-like behaviors, but no experiments were done to show that LC-specific deletion of ErbB4 recapitulates any of the effects on LC function. Furthermore, it was stated that 77% of the Cre/GFP+ neurons were TH^+^, the authors should clarify if there was off-target ErbB4 deletion. Interestingly, lithium reversed the mania-like behaviors of Th-Cre;Erbb4^loxp/loxp^ mice. However, enthusiasm was dampened for the norepinephrine and dopamine antagonist studies since the authors did not include control mice treated with the antagonists. This work is important and intriguing, but the conclusions would be better supported after some revisions and clarifications.

We thank the reviewer for these valuable suggestions. First, to test whether LC-specific ErbB4 deletion alters LC biological functions, we measured spontaneous firing, NR2B expression, and NMDA function of LC-NE neurons in viral-mediated LC-specific ErbB4 deletion mice. Results showed that similar to *Th-Cre; ErbB4^loxp/loxp^* mice, AAV-Cre-GFP-injected *ErbB4^loxp/loxp^* mice presented elevated spontaneous firing, increased NR2B expression, and enhanced NMDA receptor function (revised Figure 6).

Second, to clarify ErbB4 off-target deletion in virus-injected mice, we performed immunostaining against TH and ErbB4 on brain slices from AAV-GFP-injected *Erbb4^loxp/loxp^*mice and quantified the percentage of ErbB4-expressing (ErbB4+) cells in TH-negative (TH-) but GFP-positive (GFP+) neurons in the LC region based on co-localization of GFP, TH, and ErbB4. We’ve described in the manuscript that about 77.7% of the Cre/GFP+ neurons were TH^+^. Among the left GFP+ neurons (22.3% of total), about 55.5% were ErbB4+TH- (Figure 5—figure supplement 2). Thus, we concluded that ~10% of virus-infected neurons in AAV-injected *Erbb4^loxp/loxp^* mice may have off-target ErbB4 deletion and we’ve added relevant description in the first paragraph of the subsection “Viral-mediated LC-specific ErbB4-deficient mice recapitulates mania-like behaviors and molecular and electrophysiological changes of *Th-Cre;Erbb4^loxp/loxp^* mice”.

Last, as the reviewer pointed out, we added the groups of control mice treated with norepinephrine or dopamine antagonists and repeated behavioral tests in Figure 8. Compared with control mice, mice treated with antagonists showed decreased movement, increased anxiety and depression, and decreased sucrose preference (revised Figure 8).

The authors could quantify the immunofluorescent images for Figure 1F. At a minimum, it would be better for Figure 1F to show an image of ErbB4 in the entire LC of Th-Cre;Erbb4^loxp/loxp^ mice as opposed to a high magnification image of a few cells.

Good suggestion, we quantified the fluorescent intensity of ErbB4 in TH^+^ neurons of LC-containing brain slices from the control and *Th-Cre; Erbb4^loxp/loxp^*mice immune-stained with TH and ErbB4 antibodies. The fluorescent intensity of ErbB4 in TH^+^ neurons in *Th-Cre; Erbb4^loxp/loxp^* mice was dramatically decreased compared to that of the control mice (Figure 1G).

Please clarify whether electrophysiology experiments were done at the same time of day for control and Th-Cre;Erbb4^loxp/loxp^ mice.

All the electrophysiology experiments were done at the same time of day for control and *Th-Cre; Erbb4^loxp/loxp^* mice (performed from 13:00-17:00 consistently). We have added the description in the subsection “Electrophysiology”.

The authors should clarify what the control mice are in the electrophysiology and western blot studies. Differences in genetic background or Cre-recombinase itself could affect the reported measures.

In the electrophysiology experiment, Ai9; *Th-Cre* mice or *Erbb4^loxp/loxp^* mice injected with AAV-GFP virus were used as the controls. In Western blot analysis, controls were *ErbB4^loxp/loxp^* mice or*ErbB4^loxp/loxp^* +AAV-GFP mice. *ErbB4^loxp/loxp^* mice were littermates to the *Th-Cre; Erbb4^loxp/loxp^* mice and we have compared the behavioral performance between *Th-Cre* mice and *Erbb4^loxp/lo^*^xp^ mice and there was no significant difference between these two mouse lines (data not shown). Additionally, both the control and mutant mice were in the C57BL/6 genetic background. We have clarified that in the Materials and methods (subsections “Electrophysiology”, “Western blot analysis” and “Generation and maintenance of mice”).

Please report the number of mice used per group for the behavior experiments.

As suggested, we have described the number of mice used per group in each behavioral experiment in both the figure legend and statistical reporting file.

The authors should be careful in saying that ErbB4 was predominantly deleted from noradrenergic neurons. The authors should clarify that ErbB4 was determined to be predominantly deleted from noradrenergic neurons in the locus coeruleus in the midbrain (Figure 1). The authors did not look at ErbB4 expression in other brain regions with catecholaminergic neurons.

We thank the reviewer for the suggestion and we have made correction of relevant descriptions in the manuscript. In response to this issue, beside LC, we quantified Cre expression in our *Th-Cre* line in another two catecholaminergic neuron-enriched regions, VTA and SNC. Opposite to LC, there was minimal Cre expression (Figure 1A-B) in the VTA and SNC and therefore we pursued ErbB4 functional study in the LC. The function of ErbB4 in VTA and SNC catecholaminergic neurons is interesting and has been studied by Skirzewski et al., 2017 which we discussed in the tenth paragraph of the Discussion. ErbB4 deletion in VTA and SNC catecholaminergic neurons showed very different behavioral phenotype from our LC ErbB4-deficient mice (Skirzewski et al., 2017). Notably, we injected AAV-Cre-GFP virus specifically in LC area of *ErbB4^loxp/loxp^* mice and our results showed that region-specific deletion of ErbB4 in the LC is sufficient to induce similar electrophysiological, biochemical, and mania-like behavioral phenotypes as those manifested in *Th-Cre;Erbb4^loxp/loxp^* mice. Taken together, these results indicate that ErbB4 deletion in LC-NE neurons plays the prominent role in abnormalities of *Th-Cre;Erbb4^loxp/loxp^* mice as opposed to potential off-target ErbB4 deletions in non-LC regions (subsection “Viral-mediated LC-specific ErbB4-deficient mice recapitulates mania-like behaviors and molecular and electrophysiological changes of *Th-Cre;Erbb4^loxp/loxp^* mice”).

Please clarify how% open arm entries and% open arm time were calculated. It appears that the values were normalized to the control mice which seems unnecessary. Values were not normalized for other experiments. Typically,% open arm entries refers to the number of open arm entries relative to total arm entries expressed as a percentage, which controls for differences in overall activity.

We appreciate the reviewer’s advice and we've corrected our analysis of EPM accordingly (Figure 4H-I, Figure 5H-I, Figure 7G-H, and Figure 8F-G).

If the authors have the data it would be interesting to know whether Th-Cre;Erbb4^loxp/loxp^ mice display differences in spatial d and center entries/time in the open field.

No significant change was detected between control (*ErbB4^loxp/loxp^*) mice and *Th-Cre; ErbB4^loxp/loxp^* mice in the distance travelled in center and time spent in the center area in open field test (revised Figure 4—figure supplement 2). We added these data in the subsection “*Th-Cre;Erbb4^loxp/loxp^* mice show mania-like behaviors”. Though center time in open field test, one indicator of animal anxiety, was not consistent with another anxiety indicator, time and entries in open arms in EPM, in the ErbB4-deficient mice, the reason might be that open field test and EPM has variant sensitivity to anxiety behaviors (Carola et al., 2002; Goto et al., 1993; Laetitia Prut, 2003).

For the graphs showing inter-spike intervals, the authors may consider labeling the y-axis as "cumulative probability" as opposed to "cumulative possibility".

We thank the reviewer for pointing out this typo. We've corrected the labeling accordingly in Figure 2, Figure 3, Figure 6, and Figure 7.

The authors should describe the order of behavior testing, how many behavior tests the mice were subjected to, and the rest periods in between behavior testing.

According to the principal of beginning with the test with minimized stress stimulation, each mouse was subjected to behavioral tests in the following order: open field test, elevated plus maze test, 6-min forced swim test, and lastly, the most time-consuming sucrose preference test. For the first three tests, mice were rested in their home cage for 1-2 days between two behavior tests. For the last one, mice were rested in their home cage for 3 days before the test. The information has been added in the revised Materials and methods section (subsection “Behavioral assays”).

The authors give the time of day that the behavior tests were ran, but do not describe the 12 h light/dark schedule of the mice. Please include the lights on and off housing schedule.

Lights were on from 7:00 am-7:00 pm every day. We have added the information in the Materials and methods section (subsection “Generation and maintenance of mice”).

The authors should clarify what promoter was used for the control AAV-GFP virus.

We have added the information in the Materials and methods (subsection “Virus vectors and stereotactic injection”). The promotors for both the AAV-GFP and AAV-Cre-GFP viruses were synaptophysin promotor.